# Ada-R1: Hybrid CoT via Bi-Level Adaptive Reasoning Optimization

**Haotian Luo**[1,2]*, **Haiying He**[3]*, **Yibo Wang**[4], **Jinluan Yang**[5], **Rui Liu**[6]
**Naiqiang Tan**[6], **Xiaochun Cao**[1], **Dacheng Tao**[7], **Li Shen**[1,2]†
[1] Shenzhen Campus of Sun Yat-sen University;
[2] Center for AI Theoretical Foundation and Systems, Shenzhen Loop Area Institute;
[3] China Agricultural University; [4] Tsinghua University; [5] Zhejiang University;
[6] Didichuxing Co. Ltd; [7] Nanyang Technological University

## Abstract

Recently, long-thought reasoning models achieve strong performance on complex reasoning tasks, but often incur substantial inference overhead, making efficiency a critical concern. Our empirical analysis reveals that the benefit of using Long-CoT varies across problems: while some problems require elaborate reasoning, others show no improvement—or even degraded accuracy. This motivates adaptive reasoning strategies that tailor reasoning depth to the input. However, prior work primarily reduces redundancy within long reasoning paths, limiting exploration of more efficient strategies beyond the Long-CoT paradigm. To address this, we propose a novel two-stage framework for adaptive and efficient reasoning. First, we construct a hybrid reasoning model by merging long and short CoT models to enable diverse reasoning styles. Second, we apply bi-level preference training to guide the model to select suitable reasoning styles (group-level), and prefer concise and correct reasoning within each style group (instance-level). Experiments demonstrate that our method significantly reduces inference costs compared to other baseline approaches, while maintaining performance. Notably, on five mathematical datasets, the average length of reasoning is reduced by more than 50%, highlighting the potential of adaptive strategies to optimize reasoning efficiency in large language models. Our code is coming soon at https://github.com/StarDewXXX/AdaR1

## 1 Introduction

Recent large language models (LLMs) such as OpenAI's O1[1] and Deepseek's R1[2] adopt extended and structured reasoning processes (Long-CoT) for LLMs to enhance problem-solving, achieving strong performance through human-like deliberation. Similarly, [3] and [4] enable MLLMs to possess such capabilities as well. However, the improved reasoning comes with high inference costs, including increased latency and resource consumption[5–7], which limits deployment in real-time or resource-constrained scenarios. Existing efficiency-oriented methods operate within the Long-CoT distribution, aiming to reduce redundancy through pruning or compression[8, 9, 6]. While effective to some extent, these approaches do not question whether long reasoning is necessary, overlooking potential gains from fundamentally shorter reasoning strategies. CoT-Valve[10] enables both long and short outputs but lacks adaptive selection based on input complexity, leading to suboptimal results.

Our investigation (presented in Section 3) about the benefit of Long-CoT reasoning reveals a crucial insight: the utility of long, elaborate reasoning chains is highly problem-dependent. While complex

---

*Equal contribution
†Corresponding Author: Li Shen (shenli6@mail.sysu.edu.cn)

39th Conference on Neural Information Processing Systems (NeurIPS 2025).

problems genuinely benefit from detailed, step-by-step derivations, many other problems can be solved accurately and more efficiently with shorter, more direct reasoning paths. In fact, for simpler problems, forcing a Long-CoT process might not only be wasteful but can sometimes even introduce errors or degrade performance. This observation strongly motivates the need for adaptive reasoning strategies – systems that can tailor the depth and style of their reasoning process to the specific demands of the input problem.

Inspired by these limitations, we propose a two-stage framework for efficient and adaptive reasoning by enabling models to choose between distinct reasoning strategies. The first stage constructs a hybrid model capable of generating both Long-CoT and Short-CoT outputs. The second introduces Bi-Level Adaptive Reasoning Optimization , a training method comprising: (i) Group-Level Preference, guiding the model to select an appropriate reasoning style based on input complexity, and (ii) Instance-Level Preference, encouraging concise yet accurate reasoning within the chosen style. This dual-level adaptation allows dynamic allocation of computational resources, yielding substantial efficiency gains without sacrificing performance. On MATH[11], our method reduces reasoning length by 58% with no accuracy loss, and on GSM8K[12], by 74% with improved accuracy. These results highlight the effectiveness of adaptive reasoning in balancing quality and efficiency in large-scale models.

Our contributions can be summarized as follows:

- We conduct an empirical analysis investigating the benefits of long Chain-of-Thought (CoT) reasoning relative to shorter CoT approaches, identifying the conditions under which extended reasoning paths offer tangible advantages.
- We propose using Adaptive Hybrid Reasoning Model to enhance inference efficiency, accompanied by a novel training pipeline (Ada-R1). Comprehensive experiments demonstrate that our proposed method achieves excellent performance, significantly improving efficiency while maintaining high accuracy.
- We perform further analyses on the resulting Adaptive Hybrid Reasoning Model to gain deeper insights into its characteristics and operational behavior. And we will release the model weights of the Adaptive Hybrid Reasoning Model to the public to encourage further research and application by the community.

## 2 Related Work

**Model Merging** Model merging [13] is an emerging technique that fuses parameters from multiple trained models into one without access to original training data. Recent methods include parameter interpolation [14] and alignment-based strategies [15], with applications in LLMs, multimodal models, and other machine learning subfields. Beyond simple linear averaging, advanced methods such as DARE [16], TIES-Merging [17], and AdaMerging [18] have been proposed. DARE reduces redundancy by dropping and rescaling delta parameters. TIES-Merging mitigates interference by trimming and aligning parameter signs. AdaMerging improves performance via entropy-based layer or task weighting on unlabeled data. In contrast to traditional model merging that consolidates capabilities from multiple models, our work enables a single model to adaptively choose between Long-CoT and Short-CoT reasoning for each instance, aiming to optimize computational efficiency rather than multi-task performance.

**Efficient Reasoning** A variety of methods have been proposed for improved reasoning efficiency. Several techniques apply post-training strategies to shorten reasoning paths. [6] constructs preference datasets using DPO and SimPO, guiding models toward concise reasoning through preference-based fine-tuning. O1-Pruner[8] samples CoTs to build baselines for length and accuracy, then applies offline optimization to reduce reasoning length without harming performance. Similarly, [19] leverages simple fine-tuning on self-generated concise CoTs obtained via best-of-N sampling and few-shot prompting. Some approaches focus on token-level compression. TokenSkip[20], for instance, removes tokens selectively based on their estimated importance within the CoT. CoT-Valve[10], in contrast, manipulates the parameter space to produce CoTs with varying degrees of compression. Besides, various methods adopt different reasoning paradigms for efficiency. For instance, COCONUT[21] and CCOT[5] enable reasoning within the latent space, reducing the need for explicit token-level generation. Speculative Thinking[22] enhances small model inference by allowing large models to guide them during reasoning. Similarly, LightThinker[23] achieves efficiency by dynamically compressing intermediate thoughts throughout the reasoning process. Also,

some works ([24],[25], [26], [27], [28], [29]) design novel reasoning paradigms for efficiency. [30] also explores model merging technical for reasoning efficiency. Different from most works, our work solves reasoning efficiency in a novel adaptive reasoning perspective.

## 3 Motivation

### 3.1 Problem Setup

Chain-of-Thought (CoT) prompting has emerged as a powerful technique for enhancing the reasoning capabilities of large language models. Within the CoT paradigm, a distinction can be made between Long-CoT, which involves generating detailed and extensive thinking steps, and Short-CoT, which directly generate solving steps.

### 3.2 When Do We Need Long-CoT?

Simply applying Long-CoT to all problems introduces unnecessary overhead, especially for easier tasks where detailed reasoning brings little or no benefit. To understand when Long-CoT is truly needed, we empirically analyze its effectiveness across different problem types. We compare Long-CoT and Short-CoT on a mixed dataset (MixMathematics) composed of samples from AIME[31], MATH, and GSM8K (details in Section 5.1). We use DeepSeek-R1-Distill-Qwen-7B for Long-CoT, and fine-tune it with 2,000 Short-CoT samples from Qwen2.5-Math-7B-Instruct[32] to create a consistent Short-CoT model. We avoid using Qwen2.5 directly due to its differing training format, which may affect later merging and sampling. From 2,500 problems, we generate 12 responses per model per question and remove cases where both models fail completely. We then calculate accuracy gains (Long-CoT accuracy minus Short-CoT accuracy).

As shown in Figure 1 (left), nearly half the samples show no improvement from Long-CoT, and some even suffer performance drops. Further analysis (Figure 1, right) groups samples by the average length of their Long-CoT outputs—longer CoTs tend to correspond to harder problems. We find that Long-CoT significantly improves accuracy on complex questions but provides little or no benefit for simpler ones.

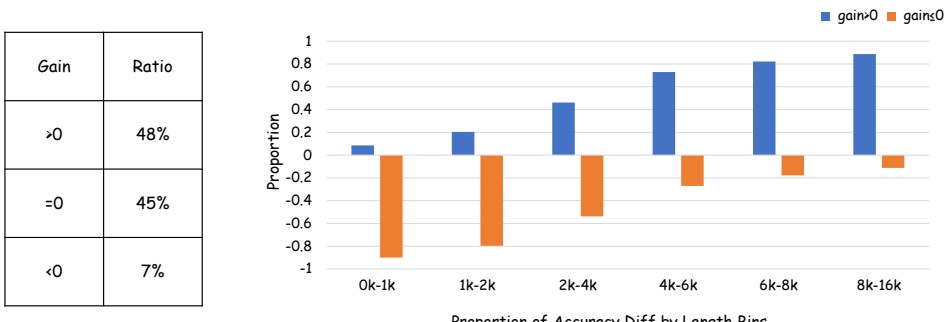

| Gain | Ratio |
| --- | --- |
| >0 | 48% |
| =0 | 45% |
| <0 | 7% |

Figure 1: The proportion of gain in the data (left) and the relationship between CoT length and accuracy improvement (right), Long-CoT reasoning improves accuracy on difficult problems but has little effect or harms performance on easy ones.

### 3.3 A New Perspective on CoT Efficiency

Prior methods (Table 1), such as Overthinking [6], kimi-1.5 [33], and O1-Pruner, typically operate within a limited optimization scope but generally maintain performance stability or incur only a slight drop, with O1-Pruner notably achieving no performance decrease. In contrast, methods designed for a broad optimization scope, including Model Merge and CoT-Valve, did not consider how to tackle easy and different problems, rendering the model incapable of determining its reasoning depth according to the inherent difficulty of the task. Thus they frequently result in significant performance degradation. In a nutshell, methods with a restricted optimization can generally preserve performance but lose the chance to utilize shorter CoT. However, approaches capable of utilize broader CoT

| Method | CoT Optimization Scope | Performance (Accuracy) |
|---|---|---|
| Overthinking[6] | Limited ✗ | Slightly Dropped ✓ |
| kimi-1.5[33] | Limited ✗ | Slightly Dropped ✓ |
| O1-Pruner | Limited ✗ | Not Dropped ✓ |
| Naive Merge | Broad ✓ | (mostly) Dropped ✗ |
| CoT-Valve | Broad ✓ | Dropped ✗ |
| Ada-R1(Ours) | Broad ✓ | Slightly Dropped ✓ |

Table 1: Comparison of Different Methods. "Limited" indicates optimization within the Long-CoT distribution, restricting efficiency. "Broader" covers both Long- and Short-CoT, enabling shorter, more efficient responses. "Slightly dropped" means accuracy decreased by less than 3%, while "dropped" refers to a decrease greater than 3%.

distribution have struggled to maintain accuracy due to their inability to adapt adequate reasoning depth to problem complexity.

The finding mentioned in last section motivates us to address the efficiency challenge of Long-CoT models from a novel perspective: enabling the reasoning model to adaptively select an appropriate reasoning mode (long or short CoT) for different problems, and then generate a correct and concise CoT in the determined mode. Our proposed method (Ada-R1) differentiates itself by successfully achieving a broad optimization scope while incurring only a marginal performance decrement. This demonstrates a more favorable trade-off between efficiency and accuracy compared to existing broad-scope optimization techniques.

## 4 Bi-Level Adaptive Reasoning Optimization

### 4.1 Problem Setup

We consider a LLM parameterized by $\theta$ and denoted as $\pi_\theta$. In the context of math problem solving, the LLM accepts a sequence $x = [x^1, \ldots, x^n]$, commonly termed as the problem, and then generate a corresponding solution $y = [y^1, \ldots, y^m]$. Hence, the solution $y$ is construed as a sample drawn from the conditional probability distribution $\pi_\theta(\cdot|x)$. The conditional probability distribution $\pi_\theta(y|x)$ can be decomposed as follows:

$$\pi_\theta(y|x) = \prod_{j=1}^{m} \pi_\theta(y^j|x, y^{<j}). \tag{1}$$

We consider two LLMs: one trained to generate long, reflective Chain-of-Thought (CoT) reasoning (*Long-CoT model*, denoted as $\theta_L$) and the other trained for short and concise reasoning paths (*Short-CoT model*, denoted as $\theta_S$). These two models are typically fine-tuned with different CoT and demonstrate distinct reasoning patterns.

### 4.2 Method Overview

Our method consists of two stages, shown in Figure 2. First, we merge a Long-CoT model and a Short-CoT model to obtain a unified reasoning model capable of generating both types of reasoning paths. This allows exploration over a broader CoT distribution. In the second stage, we apply Bi-Level Preference Training: for group-level preference, the model learns to choose between long and short reasoning group based on the input; for instance-level preference, it learns to compress the reasoning path to improve efficiency within the chosen group determined by group-level preference.

### 4.3 Stage I: Long-and-Short Reasoning Merge

To enable flexible reasoning behaviors within a single model, we first perform model merging with long and short models. We adopt a simple yet effective strategy of linearly merging their parameters. Given two models with parameters $\theta_L$ and $\theta_S$, we compute the merged model as:

$$\theta_H = \alpha\theta_L + (1 - \alpha)\theta_S, \tag{2}$$

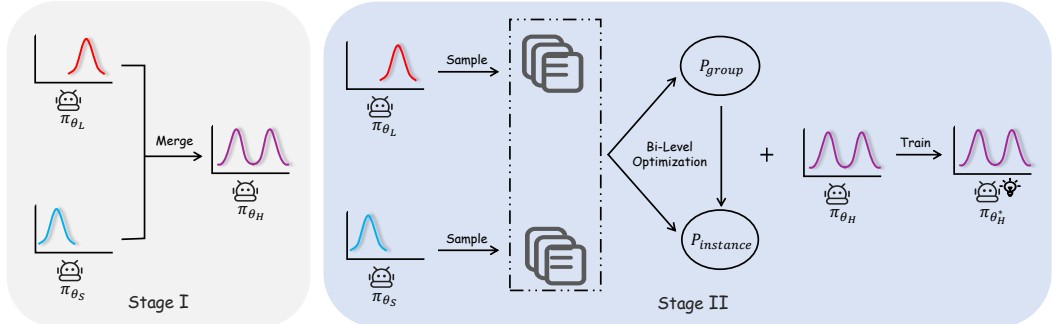

Figure 2: Pipeline of Ada-R1. At Stage I, we fused the models to obtain $\pi_{\theta_H}$. In Stage II, we sample from both long and short models and then elicit the group-level and instance-level preference. After this, we optimize $\pi_{\theta_H}$ at both group and instance level to obtain a hybrid adaptive reasoning model.

where $\alpha \in [0, 1]$ is a merging coefficient that balances the contribution from each model. The resulting hybrid reasoning model, $\pi_{\theta_H}$, inherits the capacity to generate both long and short CoT depending on the input.

This merged model expands the diversity of the CoT distribution it can produce, laying the foundation for adaptive reasoning. By combining the strengths of both reasoning styles, it enables the model to potentially match different problem types with suitable reasoning strategies, which is key to improving efficiency in the next stage.

### 4.4 Stage II: Bi-Level Preference Training

In this stage, we introduce a Bi-Level Preference Training strategy to fine-tune the model toward efficient reasoning. The core idea is to train the model to: (1) select the appropriate reasoning style (long or short) for each problem (*group-level preference*) and (2) further compress the reasoning within the determined chosen group (*instance-level preference*).

**Group Labels.** We define a group label $g$ to denote the reasoning style of a response group. Let $g_L$ denote the *long reasoning group* and $g_S$ denote the *short reasoning group*. For a given input problem $x$, a generated resposne (solution) $y$ belongs to one of the two groups. We use $\{y_i\}_{g=g_L}$ to denote the set of $K$ Long-CoT responses generated by the Long-CoT model $\theta_L$, and $\{y_j\}_{g=g_S}$ for the corresponding short responses from the Short-CoT model $\theta_S$.

**Group-Level Preference.** For each math problem $x$ in the dataset $\mathcal{D}$, we sample $K$ solutions from both the long and short reasoning models. Let $\{y_i^L\}_{i=1}^K$ and $\{y_j^S\}_{j=1}^K$ be the respective sample sets. We define the approximated accuracy expectation for each group as:

$$\hat{\mathbb{E}}[C^L(x)] = \frac{1}{K}\sum_{i=1}^K \mathbb{1}[\text{Correct}(y_i^L)], \quad \hat{\mathbb{E}}[C^S(x)] = \frac{1}{K}\sum_{j=1}^K \mathbb{1}[\text{Correct}(y_j^S)], \tag{3}$$

where $\mathbb{1}[\cdot]$ is the indicator function. Then we introduce a preference margin threshold $\epsilon > 0$. The group-level preference for $x$ is then determined as:

$$\begin{cases} g_L \succ g_S \mid x & \text{if} \quad \hat{\mathbb{E}}[C^L(x)] - \hat{\mathbb{E}}[C^S(x)] > \epsilon, \\ g_S \succ g_L \mid x & \text{if} \quad \hat{\mathbb{E}}[C^L(x)] - \hat{\mathbb{E}}[C^S(x)] \le \epsilon. \end{cases}$$

Given the group-level preference for an input $x$, we form training pairs from the Cartesian product of the two groups. For example, if $g_L \succ g_S \mid x$, we construct the preference pairs as:

$$\mathcal{P}_{\text{group}}(x) = \left\{ (x, y_i^L, y_j^S) \mid i \in [1, K], j \in [1, K] \right\}. \tag{4}$$

From this set of pairs, we randomly sample a subset contain $M_1$ pairs to construct DPO training tuples $(x, y_w, y_l)$, where $y_w$ is the preferred (chosen) response and $y_l$ is the less preferred (rejected).

For all $x \in \mathcal{D}$, we perform group-level preference assignment by comparing the sampled long and short responses as described above. These tuples are then aggregated into a new dataset $\mathcal{D}_{\text{group}} = \{(x, y_w, y_l)\}$, which serves as supervision for optimizing the DPO objective at the group level.

**Instance-Level Preference.** Once the preferred group $g^* \in \{g_L, g_S\}$ is determined for a given $x$, we further construct *instance-level preferences* within that group to encourage more concise reasoning. We compare response pairs $(y_a, y_b)$ such that both belong to the same group (e.g., $y_a, y_b \in \{y_i^L\}$), and prefer the shortest correct response. For dispreferred samples, we select $M_2$ longest responses. Formally, for each $x \in \mathcal{D}$ with preferred group $g^*$, we first identify the subset of correct responses $\{y_i\}_{\text{correct}} \subseteq \{y_i\}_{g=g^*}$. Among these, we select the shortest correct response as the preferred instance:

$$y_w = \arg \min_{y \in \{y_i\}_{\text{correct}}} |y|.$$

To construct instance-level preference pairs, we then select the $M_2$ longest responses from the entire group $\{y_i\}_{g=g^*}$. Denote these as $\{y_{l_j}\}_{j=1}^{M_2}$. This yields a dataset of instance-level training tuples:

$$\mathcal{D}_{\text{instance}} = \left\{ (x, y_w, y_l) \,\middle|\, y_w = \arg\min_{y \in \{y_i\}_{g=g^*}^{\text{correct}}} |y|, y_l \in \arg\max_{y \in \{y_i\}_{g=g^*}}^{(M_2)} |y| \right\}$$

These instance-level preferences encourage the model not only to reason correctly, but also to do so concisely within the preferred reasoning style.

We sample such intra-group pairs and use them as additional training data for DPO to encourage the model to favor more concise reasoning within each group.

**Objective.** Given collelcted preference datasets $\mathcal{D}_{\text{group}}$ and $\mathcal{D}_{\text{instance}}$ sampled from $p^*$ which contains $N$ preference pairs $(x, y_w, y_l)$. With a parameter $\beta$ controlling the deviation from the reference model $p_{\text{ref}}$, DPO optimize the model by:

$$\max_{\pi_{\theta_H}} \mathbb{E}_{(x, y_w, y_l) \sim \mathcal{D}_{\text{group}} \cup \mathcal{D}_{\text{instance}}} \left[ \log \sigma \left( \beta \log \frac{\pi_{\theta_H}(y_w \mid x)}{\pi_{\theta_{ref}}(y_w \mid x)} - \beta \log \frac{\pi_{\theta_H}(y_l \mid x)}{\pi_{\theta_{ref}}(y_l \mid x)} \right) \right]$$

## 5 Experiments

### 5.1 Setup

**Long-CoT Models.** The long thought models we chosen for our experiment are DeepSeek-R1-Distill-Qwen-7B and DeepSeek-R1-Distill-Qwen-1.5B, which have demonstrated excellent performance on most math problem-solving tasks. For both models, we utilize full-parameter fine-tuning.

**Short CoT Models.** Since model merging requires Shot-CoT models, we face two issues with existing Shot-CoT models: (1) they often employ templates that differ from those used in Long-CoT models; (2) they tend to exhibit substantial parameter deviations from the base model, which introduces instability during the merging process[18, 34]. To address these challenges, we fine-tune the Long-CoT models using a small number of short CoT examples to obtain the corresponding Shot-CoT models. This approach ensures consistency in template usage and maintains a closer parameter proximity between the two models.

**Dataset.** Following s1[35] and Light-R1[36], we construct a mixed training dataset to ensure coverage across mathematical problems of varying difficulty levels. Specifically, we combine GSM8K, MATH, and AIME datasets in a ratio of 1:3:1, resulting in a total of 2,500 diverse math problems.

**Evaluation.** We use the GSM8K test set, the MATH test set, and AIME25 as in-distribution evaluation data, while Olympiad[37] and Minerva[38] are employed as out-of-distribution test sets. For evaluation metrics, we consider both accuracy and sequence length. For small-scale datasets such as AIME25, we report results averaged over four independent runs to reduce randomness and improve reliability. Additionally, we report the average accuracy degrade rate and the average length reduction rate across all test sets.

Table 2: Accuracy (shown above) and length (shown below) of models and methods on different benchmarks. Avg represents the change in length and accuracy compared to the Long model (+ for increase, - for decrease).

| Model / Bench | AIME25 | MATH500 | GSM8K | Olympiad | Minerva | Avg.(%) |
|---|---|---|---|---|---|---|
| **7B Models** | | | | | | |
| Long(R1-distill) | 38.3 | 90.2 | 88.9 | 54.4 | 35.7 | - |
| | (11005) | (3534) | (1014) | (7492) | (4533) | - |
| Short | 10.0 | 78.6 | 89.5 | 39.4 | 28.6 | −19.97% |
| | (957) | (591) | (272) | (910) | (579) | (-84.57%) |
| Merge | 21.7 | 79.4 | 88.4 | 41.2 | 25.7 | −18.63% |
| | (9079) | (916) | (236) | (3743) | (1734) | (-56.02%) |
| DPO | 35.8 | 89.4 | 86.0 | 55.2 | 35.6 | -3.56% |
| | (9976) | (2334) | (360) | (5309) | (3281) | (-33.26%) |
| O1-Pruner | 40.0 | 92.4 | 89.4 | 55.3 | 35.3 | +2.48% |
| | (9353) | (2212) | (377) | (5295) | (3259) | (-34.53%) |
| CoT-Valve | 22.5 | 78.6 | 87.9 | 39.6 | 29.4 | −18.41% |
| | (5024) | (747) | (235) | (2313) | (629) | (-73.06%) |
| Ada-R1(Ours) | 35.8 | 90.2 | 90.3 | 52.4 | 34.1 | -1.65% |
| | (8426) | (1468) | (260) | (4889) | (1647) | (-50.93%) |
| **1.5B Models** | | | | | | |
| Long(R1-distill) | 23.3 | 81.0 | 80.9 | 41.6 | 26.1 | - |
| | (12307) | (4416) | (1481) | (7687) | (5789) | - |
| Short | 9.0 | 69.4 | 78.2 | 30.7 | 22.4 | −26.34% |
| | (1098) | (740) | (269) | (1373) | (725) | (-85.15%) |
| Merge | 20.8 | 71.8 | 74.2 | 28.6 | 20.0 | −10.12% |
| | (9226) | (1740) | (251) | (3767) | (1399) | (-59.10%) |
| DPO | 20.8 | 81.4 | 74.8 | 42.8 | 24.3 | −5.93% |
| | (10224) | (3055) | (374) | (6319) | (3905) | (-34.57%) |
| O1-Pruner | 23.3 | 82.6 | 84.6 | 44.7 | 28.3 | +2.18% |
| | (9496) | (2782) | (726) | (5658) | (3964) | (-33.75%) |
| CoT-Valve | 14.2 | 69.6 | 76.3 | 28.7 | 19.5 | −19.61% |
| | (7744) | (1299) | (205) | (3169) | (867) | (-67.52%) |
| Ada-R1(Ours) | 23.0 | 80.8 | 79.2 | 42.1 | 23.5 | -1.21% |
| | (9516) | (2455) | (341) | (5802) | (3021) | (-43.28%) |

## 5.2 Competitive Methods

**DPO.** DPO are widely used baselines in reasoning optimization area. Follwoing the setting of [33, 6], we choose shortest sample as chosen samples and longest sample as rejected sample.

**CoT-Valve.** CoT-Valve enables dynamic control of Chain-of-Thought length using a single model by identifying and leveraging a controllable direction in the model's parameter space to generate compressed CoT.

**O1-Pruner.** O1-Pruner is a method designed to reduce reasoning overhead while maintaining model accuracy. It begins by establishing a baseline through pre-sampling, and then applies reinforcement learning-based finetuning.

## 5.3 Main Results

We can be seen from the Table 2 that: the Short and Merge models achieve the most significant length reduction compared to the Long Model. However, this efficiency gain is accompanied by a notable degradation in accuracy, exceeding 10 percentage points. Among the models that do not suffer significant accuracy degradation, our method achieves the best length reduction performance,

Table 3: Ablation study of each component on several benchmarks, showing that the Merge + bi-level achieves the best trade-off, with a 52.08% average length reduction and a minimal 0.51% accuracy degradation compared to others.

| Bench / Model | AIME25 | MATH500 | GSM8K | Avg.(%) |
|---|---|---|---|---|
| Long(R1-distill) | 38.3 | 90.2 | 88.9 | - |
| | (11005) | (3534) | (1014) | - |
| Merge | 21.7 | 79.4 | 88.4 | −12.83% |
| | (9079) | (916) | (236) | (-56.10%) |
| Merge + SFT | 35.8 | 84.6 | 88.7 | -3.82% |
| | (11222) | (2314) | (375) | (-31.86%) |
| Merge + instance level | 24.2 | 81.6 | 88.0 | −10.86% |
| | (8514) | (886) | (212) | (-38.20%) |
| Merge + group level | 30.8 | 87.8 | 91.6 | -3.31% |
| | (9049) | (1565) | (359) | (-46.03%) |
| Merge + bi level | 35.8 | 90.2 | 90.3 | -0.51% |
| | (8426) | (1468) | (260) | (-52.08%) |

reaching 50.93% for the 7B model and 43.28% for the 1.5B model. Compared to DPO, our approach demonstrates both more substantial length reduction and significantly less accuracy degradation. While O1-Pruner maintains high accuracy, its length reduction effect is considerably weaker than that of our method.

## 5.4 Ablation Study

To assess each component's impact in our framework, we conduct an ablation study on AIME25, MATH500, and GSM8K. As shown in Table 3, the Merge model reduces average output length by 56.10%, but with a notable 12.83% drop in accuracy.

Supervised Fine-Tuning (SFT) on the merged model (using the chosen sample in our group level preference dataset), helps recover a significant portion of the lost accuracy, bringing the average degradation down to 3.82%. However, its average length reduction is less pronounced (31.86%) compared to the Merge model without further training.

Although optimization at the instance level leads to a substantial reduction in sequence length (38%), the model fails to distinguish which problems require long reasoning and which can be solved with shorter responses, resulting in more than a 10% drop in accuracy.

Introducing the group-level preference training after merging (Merge + group level) yields better results than SFT and only instance level. It achieves a higher average length reduction (46.03%) and a slightly better accuracy recovery, with only a 3.31% average degradation relative to the baseline. This indicates that training the model to select the appropriate reasoning style is effective in balancing efficiency and accuracy.

The full method (Merge + bi level), combining group and instance level preference training, offers the best trade-off: 52.08% length reduction with only 0.51% accuracy loss. This result highlights the complementary benefits of the bi-level training approach: the group level guides the model towards suitable reasoning styles, and the instance level further refines the chosen style by favoring concise and correct responses, leading to a highly efficient and accurate hybrid reasoning model.

# 6 Further Evaluation

## 6.1 Thinking Ratio Study

To investigate the thinking characteristics of different models, we propose the "Thinking Ratio" metric. This metric is designed to detect whether a response constitutes a deep thinking (Long-CoT) sample. Long-CoT responses typically include unique keywords (e.g., 'wait', 'recheck'). By detecting the

presence of these keywords in a response, we can determine if it is a deep thinking sample. This detection method is more generalizable than relying solely on response length. We use a subset of Math Testset. Using the method described above, we analyzed the proportion of deep thinking samples for each model. Furthermore, for each category (thinking/non-thinking samples), we also calculated their accuracy.

The results are shown in Figure 3. The baseline Long-CoT model predominantly employs deep thinking (0.98), yielding high accuracy. In contrast, the Naive Merge model drastically shifts towards non-thinking responses (0.94) but suffers significant accuracy degradation on both thinking (0.68) and non-thinking (0.81) paths. DPO shows a moderate shift to non-thinking (0.34) while preserving accuracy. Our Ada-R1 model achieves a more significant shift towards non-thinking (0.72) than DPO, yet crucially maintains high accuracy for these dominant non-thinking responses (0.96), unlike the Naive Merge. This demonstrates Ada-R1's effective adaptation, utilizing efficient shorter paths without substantial accuracy loss.

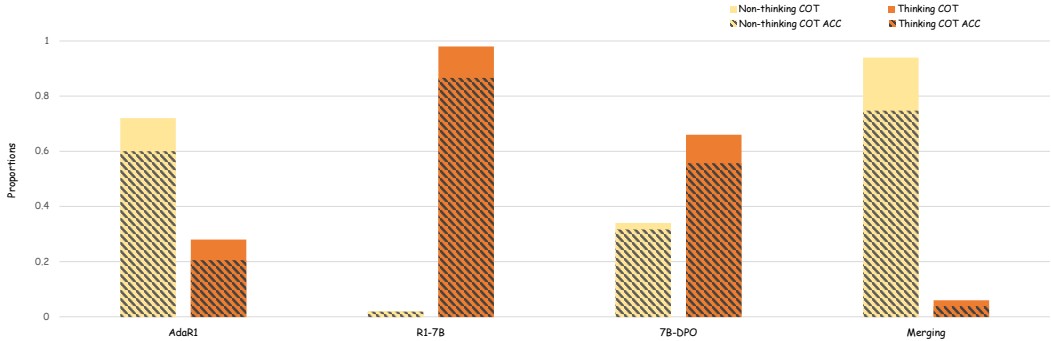

Figure 3: The proportion and accuracy of thinking and non-thinking in different methods, Ada-R1 can achieve a good balance and accuracy between thinking and non-thinking.

## 6.2 Adaptive Reasoning Study

This section evaluates the adaptive reasoning ability of Ada-R1 (7B) on the MATH dataset, which is divided into five difficulty levels (Level 1–5). We analyze both the model's thinking ratio (Long-CoT usage) and its average accuracy across these levels. As shown in the left part of Figure 4, the thinking ratio increases significantly with task difficulty. Level 1 problems have the lowest Long-CoT usage, while Level 5 shows the highest, indicating that Ada-R1 adaptively chooses to think more on harder problems. In terms of accuracy (Figure 4, right), Ada-R1 achieves strong performance across difficulty levels. Its accuracy is comparable to that of a full Long-CoT model (Deepseek-R1-Qwen-7B-Distill) and consistently higher than the Short-CoT model, especially on Levels 3 to 5. These results support our hypothesis from Section 3: Ada-R1 can selectively apply Long-CoT when needed, achieving a better balance between accuracy and efficiency.

## 6.3 Merge Model Analysis

To verify the effectiveness of the model merging strategy, we compute the average loss of the merged model on both Long-CoT and Short-CoT samples, and compare these values with those of the original Long-CoT and Short-CoT models. As shown in Table 4, the merged model exhibits similar losses to the corresponding original models on both types of samples. This indicates that the merged model preserves the reasoning capabilities of both individual models, thereby enabling it to fully leverage population-level preference signals during the second-stage fine-tuning. These results support the effectiveness of our dual-level preference training setup, where the merged model benefits from a broader reasoning space while being guided by meaningful preference supervision.

## 6.4 Out of Domain Test

To further assess the effectiveness of our method, we additionally evaluate it on non-mathematical reasoning benchmarks, including LogiQA[39], GPQA[40], and MMLU[41](we report results on both

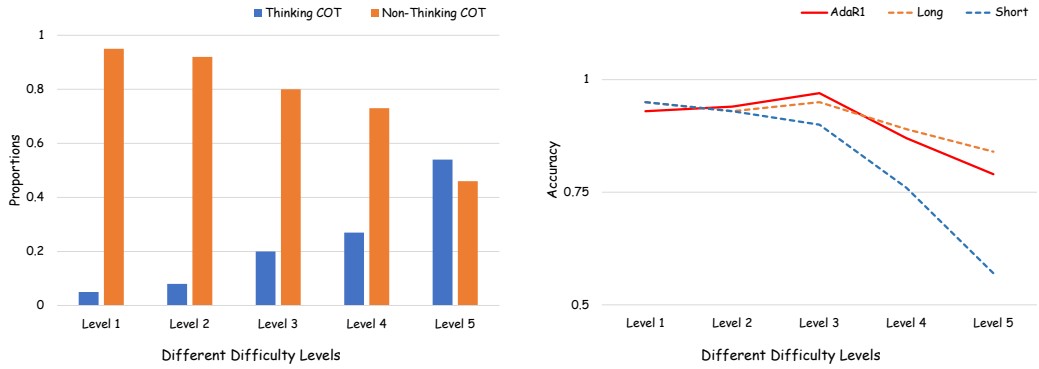

Figure 4: The ratio of thinking and non-thinking CoTs of Ada-R1-7B on different MATH levels (left) and the accuracy on different MATH levels of different models (right). As the difficulty increases, Ada-R1 is able to think more on harder problems and maintain higher accuracy.

Table 4: Average losses of the merged model compared with the original Long-CoT and Short-CoT models, showing that the merged model retains both reasoning abilities.

| Model | Long-CoT Loss | Short-CoT Loss |
|---|---|---|
| Long (R1-distill) | 0.2106 | – |
| Short | – | 0.1570 |
| Merge | 0.2141 | 0.2028 |

Table 5: Evaluation results of Ada-R1 on non-mathematical reasoning tasks.

| Bench
Model | LogiQA | GPQA | MMLU(stem) | MMLU(humanities) | Avg.(%) |
|---|---|---|---|---|---|
| Long(R1-distill) | 45.1 | 30.3 | 43.3 | 16.3 | - |
| | (3269) | (7302) | (2336) | (1476) | - |
| DPO | 44.2 | 31.3 | 48.3 | 21.0 | +7.26% |
| | (2650) | (6592) | (1577) | (1270) | (-15.95%) |
| Ada-R1(ours) | 48.6 | 32.8 | 54.6 | 30.9 | +23.63% |
| | (2794) | (5101) | (1909) | (930) | (-25.37%) |

STEM and Humanities subsets), as shown in Table 5. The results indicate that Ada-R1 maintains strong efficiency–accuracy trade-offs beyond purely mathematical problems, demonstrating its generalizability to broader reasoning domains.

# 7 Conclusion

In this paper, we demonstrate through empirical analysis that the benefits of Long-CoT reasoning vary significantly depending on the problem. Motivated by this, we propose a novel two-stage training framework for adaptive reasoning. Experiments show that model trained with our method can reason adaptively to different problems. And our method significantly reduces inference costs while preserving performance, highlighting the promise of adaptive strategies for optimizing reasoning efficiency in large language models.

# 8 Acknowledgement

This work is supported by National Key R&D Projects (NO. 2024YFC3307100), NSFC Grant (No. 62576364), National Natural Science Foundation of China (No.62025604, 62411540034), Shenzhen Basic Research Project (Natural Science Foundation) Basic Research Key Project (NO. JCYJ20241202124430041), CCF-DiDi GAIA Collaborative Research Funds (NO. CCF-DiDi GAIA 202419 and CCF-DiDi GAIA 202519).

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

# A    Selection of $\alpha$ in Stage I (Model Merge)

To select an appropriate value of $\alpha$ during the merge phase, we adopted the following approach. We randomly sampled 100 problems from the AIME exams (AIME 2025 is excluded). Using the 7B model as mentioned in our experiment, we evaluated performance under different values of $\alpha$ (0.9, 0.8, 0.7), computing both accuracy and thinking ratio for each setting. We selected $\alpha = 0.8$ as a balanced choice for Stage II training as it has relatively high accuracy and moderate thinking ratio.

Table 6: Performance of merged 7B models on 100 AIME problems.

| $\alpha$ | Accuracy | Thinking Ratio |
|---|---|---|
| 0.9 | 54.0 | 93% |
| 0.8 | 40.0 | 48% |
| 0.7 | 27.0 | 9% |

# B    Training Details

For both models, we selected 2,500 problems from the mixed Mathematics as training data. For each problem, we sample 12 times. From each set of solutions, we randomly selected 2 solutions for training. After computing the rewards, we normalized the reward values. Both models are trained with 8 * A800-80G GPUs. The other hyperparameters used in the training process are presented in the table below.

Table 7: Hyperparameters for the Deepseek-Distill-1.5B and Deepseek-Distill-7B.

| Hyperparameter | Deepseek-Distill-1.5B | Deepseek-Distill-7B. |
|---|---|---|
| cutoff_len | 4096 | 4096 |
| batch_size | 32 | 32 |
| learning_rate | 5.0e-7 | 5.0e-7 |
| num_train_epochs | 2.0 | 2.0 |
| lr_scheduler_type | constant | constant |
| $M_1$ | 4 | 4 |
| $M_2$ | 2 | 2 |
| beta | 0.05 | 0.1 |

# C    Further Evaluation of Different Methods

We further evaluate the performance and efficiency of different methods (Ada-R1, DPO, O1-Pruner) across varying levels of problem difficulty, as illustrated in Figure 5 and Figure 6. Figure 5 presents the accuracy ratio of each method relative to a baseline model across different difficulty levels within the MATH dataset. The results indicate that while performance trends may vary, our proposed Ada-R1 method demonstrates strong robustness. Specifically, as the inherent difficulty of the mathematical problems increases, Ada-R1 is able to consistently maintain a high accuracy ratio.

Figure 6 show the ratio of average tokens consumed by each method to solve problems across the same difficulty spectrum. As expected, solving more difficult problems generally requires more reasoning steps and thus more tokens. However, Figure 6 reveals that Ada-R1 exhibits favorable token efficiency. Critically, when faced with increasing problem difficulty, Ada-R1 manages to solve these complex problems while utilizing relatively fewer tokens compared to other evaluated methods, showcasing its ability to achieve efficient reasoning even for demanding tasks.

Collectively, these figures highlight Ada-R1's ability to strike a beneficial balance between accuracy and efficiency. It not only maintains high performance on challenging problems (Figure 5) but also does so in a computationally efficient manner, particularly evident in its lower token usage for difficult instances (Figure 6), addressing limitations observed in prior methods.

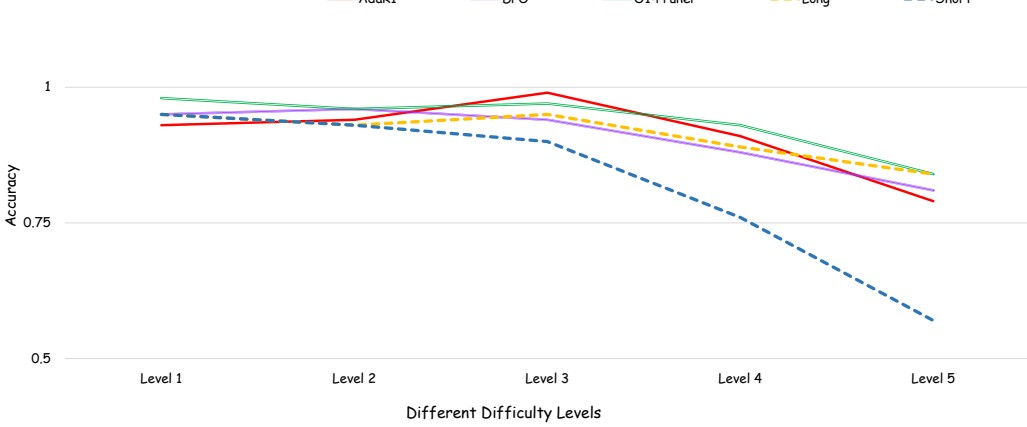

Figure 5: The ratio of accuracy at different MATH levels on different models. As the difficulty increases, Ada-R1 is able to maintain high accuracy.

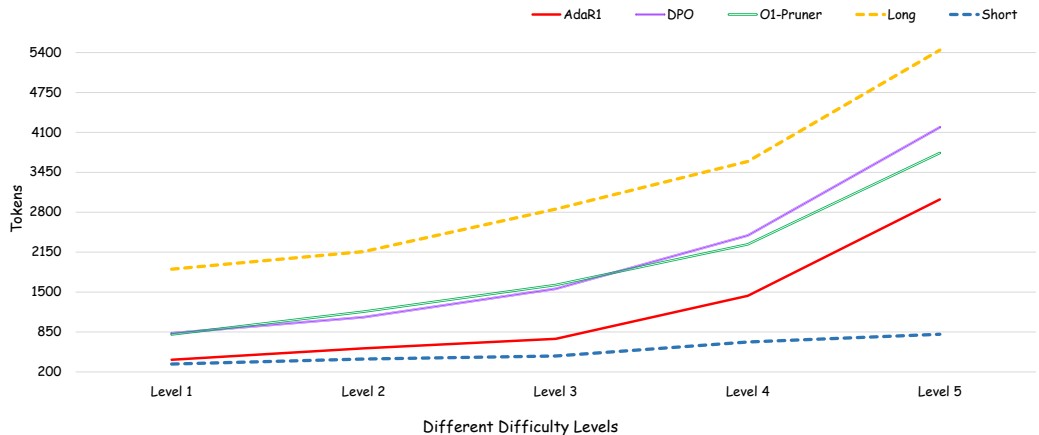

Figure 6: The ratio of average tokens on different models. As the difficulty increases, Ada-R1 is able to use relatively fewer tokens to solve difficult problems.

## D    Why Does Ada-R1 Work?

### D.1    Early Mode Selection Assumption

While Ada-R1 significantly reduces inference cost by adaptively selecting a reasoning strategy during the inference stage, its design relies on an important assumption: the model determines the reasoning mode (Long-CoT or Short-CoT) immediately after receiving the problem input, without relying on any intermediate computation or external signals. In other words, the model is expected to assess the complexity of the problem and select an appropriate reasoning path before beginning the actual problem-solving process.

### D.2    Visualization Setup

To investigate this question and better understand how Ada-R1 works, we design an experiment. We randomly select 500 problems from the training data and evaluate them using the 7B models (R1, and Ada-R1). For each problem, we extract the hidden states of the final token in the input sequence and use the last layer's hidden states as the internal representation of the problem. Based on previously computed group-level preferences (i.e., whether the problem should be solved using Long-CoT or Short-CoT), we assign a color label to each sample—red for problems requiring Long-CoT and blue

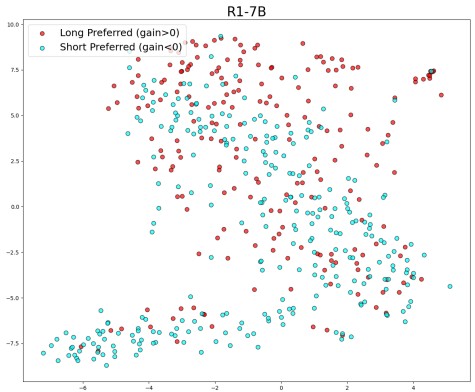

Figure 7: Visualization of R1 model

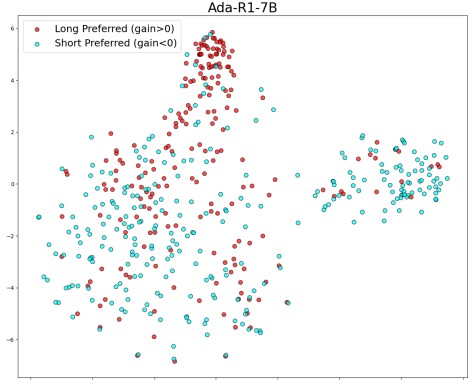

Figure 8: Visualization of Ada-R1 model

for those suitable for Short-CoT. We then apply t-SNE to project the high-dimensional hidden states into a two-dimensional space for visualization.

### D.3 Ada-R1 Learns an Implicit Problem Classifier

From the visualization, we observe that after preference-based training, Ada-R1 is able to partially separate problems that require Long-CoT from those that do not, based solely on their internal representations. This suggests that the model learns to encode problem complexity in representation space, enabling early and efficient reasoning mode selection. Such a capability underpins the effectiveness of Ada-R1: by making an informed decision on the reasoning strategy at the problem stage, the model avoids unnecessary computation for simpler problems while retaining full reasoning capacity for more complex ones.

## E   Case Study

We select two cases (9 and 10) of R1 and Ada-R1 to better understand our proposed method. In case 9, Ada-R1 can infer answers to simple problems with a small amount of CoT; in case 10, Ada-R1 can infer answers to difficult problems with relatively few long CoT.

## F   Limitations

While our proposed adaptive hybrid reasoning framework shows strong performance on benchmark datasets, its applicability in real-world scenarios presents new challenges. In particular, real-world tasks often involve diverse input distributions, domain-specific knowledge, and evolving requirements that differ from curated academic datasets such as MATH or GSM8K. The current design assumes clear patterns in problem complexity that can guide reasoning strategy selection; however, in practical applications, this complexity may be harder to estimate reliably without additional context or metadata. We view this as a promising direction for future research to bridge the gap between controlled evaluation settings and real-world reasoning applications.

**Question: Convert the point** $(0, 3)$ **in rectangular coordinates to polar coordinates. Enter your answer in the form** $(r, \theta)$**, where** $r > 0$ **and** $0 \leq \theta < 2\pi$**. Let's think step by step and output the final answer within boxed{}. Ground Truth:** $(3, \frac{\pi}{2})$

**Ada-R1**: To convert the rectangular coordinate $(0, 3)$ to polar coordinates, we use the formulas: $r = \sqrt{x^2 + y^2}$ and $\theta = \arctan\left(\frac{y}{x}\right)$. For the point $(0, 3)$: - $r = \sqrt{0^2 + 3^2} = \sqrt{9} = 3$ - Since the point lies on the positive y-axis, the angle $\theta$ is $\frac{\pi}{2}$. Thus, the polar coordinates are $(3, \frac{\pi}{2})$. Final Answer: $\boxed{(3, \dfrac{\pi}{2})}$ ✓

**R1**: Okay, so I need to convert the rectangular coordinate $(0, 3)$ to polar coordinates. Hmm, polar coordinates are represented as $(r, \theta)$, where $r$ is the radius (distance from the origin) and $\theta$ is the angle made with the positive x-axis. Let me recall the formulas to convert from rectangular to polar coordinates.

First, I remember that the radius $r$ can be found using the Pythagorean theorem. Since in rectangular coordinates, the point is $(x, y)$, the distance from the origin is $r = \sqrt{x^2 + y^2}$. So in this case, $x$ is 0 and $y$ is 3. Plugging that into the formula, $r$ should be $\sqrt{0^2 + 3^2}$ which is $\sqrt{0 + 9} = \sqrt{9} = 3$. Okay, that seems straightforward.

Now, the angle $\theta$. I know that $\theta$ is calculated using the arctangent function because $\tan \theta = \frac{y}{x}$. But wait, in this case, $x$ is 0. Hmm, arctangent of $\frac{y}{0}$. But dividing by zero is undefined. I need to think about what angle $\theta$ corresponds to when $x$ is 0.

In rectangular coordinates, when $x$ is 0, the point lies on the y-axis. Since $y$ is positive here ($y = 3$), the point is on the positive y-axis. So, in terms of polar coordinates, the angle $\theta$ should be the angle that points directly upwards along the y-axis. I think that's $\frac{\pi}{2}$ radians, which is 90 degrees. Because starting from the positive x-axis, if we rotate $\frac{\pi}{2}$ radians counterclockwise, we end up pointing straight up along the positive y-axis.

Let me double-check that. If $\theta$ is $\frac{\pi}{2}$, then converting back to rectangular coordinates, $x$ should be $r \cos \theta = 3 \cos(\frac{\pi}{2})$ and $y$ should be $r \sin \theta = 3 \sin(\frac{\pi}{2})$. $\cos(\frac{\pi}{2})$ is 0 and $\sin(\frac{\pi}{2})$ is 1, so that gives $x = 0$ and $y = 3$, which is exactly the point we started with. So that makes sense. Wait, but could $\theta$ also be $\frac{3\pi}{2}$? No, because $\frac{3\pi}{2}$ is pointing downward along the negative y-axis, which would give us $y = -3$, not 3. So $\theta$ has to be in the range where the point is on the positive y-axis, which is $\frac{\pi}{2}$.

Just to make sure, let's think about the unit circle. At $\theta = 0$, we're on the positive x-axis; at $\theta = \frac{\pi}{2}$, positive y-axis; at $\theta = \pi$, negative x-axis; at $\theta = \frac{3\pi}{2}$, negative y-axis. So yeah, $\theta = \frac{\pi}{2}$ is correct for the positive y-axis.

So, putting it all together, $r$ is 3 and $\theta$ is $\frac{\pi}{2}$. Therefore, the polar coordinates should be $(3, \frac{\pi}{2})$. Let me just write that out to make sure I didn't make any mistakes.

Wait, another thought: sometimes people use different ranges for $\theta$, like from $-\pi$ to $\pi$. But in this case, the problem specifies that $\theta$ should be between 0 and $2\pi$, so $\frac{\pi}{2}$ is within that range. So, I don't need to adjust $\theta$ or anything.

Just to recap: the point is $(0, 3)$ in rectangular, so $x = 0$, $y = 3$. Calculating $r$: $\sqrt{0^2 + 3^2} = 3$. Calculating $\theta$: since $x = 0$ and $y$ is positive, $\theta$ is $\frac{\pi}{2}$. So, $(3, \frac{\pi}{2})$ is the polar coordinate.

I think that's solid. Let me visualize it too. If I plot $(0, 3)$ on the rectangular coordinate system, it's on the y-axis, three units up. In polar coordinates, that's a radius of 3 and an angle of $\frac{\pi}{2}$ from the positive x-axis. Yep, that lines up perfectly.

Just to add another check, let me compute the sine and cosine with $\theta = \frac{\pi}{2}$. $\cos(\frac{\pi}{2})$ is 0, so $x = 3 * 0 = 0$, which matches. $\sin(\frac{\pi}{2})$ is 1, so $y = 3 * 1 = 3$, which also matches. Perfect, no discrepancies there.

So, I think I'm confident that the polar coordinates are $(3, \frac{\pi}{2})$. I don't see any mistakes in my reasoning, and all the checks seem to confirm that this is the correct answer.

Final Answer: The polar coordinates are $\boxed{(3, \dfrac{\pi}{2})}$. ✓

Figure 9: Ada-R1 can infer answers to simple problems with a small amount of CoT, whereas the R1 model requires a long CoT.

**Question:** Define $p = \sum_{k=1}^{\infty} \frac{1}{k^2}$ and $q = \sum_{k=1}^{\infty} \frac{1}{k^3}$. **Find a way to write** $\sum_{j=1}^{\infty} \sum_{k=1}^{\infty} \frac{1}{(j+k)^3}$ **in terms of** $p$ **and** $q$. **Let's think step by step and output the final answer within boxed{}. Ground Truth:** $p - q$

**Ada-R1**: Okay, so I have this problem where I need to express a double sum in terms of two known series, p and q. Let me first write down what's given. We have: $p = \sum_{k=1}^{\infty} \frac{1}{k^2}$ and $q = \sum_{k=1}^{\infty} \frac{1}{k^3}$. And the expression we need to evaluate is $\sum_{j=1}^{\infty} \sum_{k=1}^{\infty} \frac{1}{(j+k)^3}$. Hmm, so this is a double sum over j and k starting from 1 to infinity, and each term is 1 divided by $(j+k)$ cubed. I need to find a way to write this in terms of p and q. Let me think about how to approach this. Maybe I can change variables to simplify the expression. If I let $n = j + k$, then for each fixed n, j can range from 1 to n-1, and k would be n - j. So, for each n, the number of terms where j + k = n is n - 1. So, maybe I can rewrite the double sum as a single sum over n, and for each n, sum over j from 1 to n-1. Let me write that down: $\sum_{j=1}^{\infty} \sum_{k=1}^{\infty} \frac{1}{(j+k)^3} = \sum_{n=2}^{\infty} \sum_{j=1}^{n-1} \frac{1}{n^3}$. Yes, that seems right. Because when j + k = n, j can go from 1 to n-1, and for each j, k is determined as n - j. So, the inner sum is just adding $\frac{1}{n^3}$ (n-1) times. So, the double sum becomes: $\sum_{n=2}^{\infty} \frac{n-1}{n^3}$. Simplify that expression: $\sum_{n=2}^{\infty} \frac{n-1}{n^3} = \sum_{n=2}^{\infty} \left( \frac{1}{n^2} - \frac{1}{n^3} \right)$. Yes, because $\frac{n-1}{n^3} = \frac{n}{n^3} - \frac{1}{n^3} = \frac{1}{n^2} - \frac{1}{n^3}$. So now, the sum can be split into two separate sums: $\sum_{n=2}^{\infty} \frac{1}{n^2} - \sum_{n=2}^{\infty} \frac{1}{n^3}$. But notice that these sums are just p and q minus their first term. Let's write that: The first sum is: $\sum_{n=2}^{\infty} \frac{1}{n^2} = p - \frac{1}{1^2} = p - 1$. Similarly, the second sum is: $\sum_{n=2}^{\infty} \frac{1}{n^3} = q - \frac{1}{1^3} = q - 1$. Therefore, the original ...
**[OMITTED]**
Thus, the original double sum becomes: $(p-1) - (q-1) = p - 1 - q + 1 = p - q$. Therefore, the final answer is: $\boxed{p - q}$. ✓

**R1**: Okay, so I have this problem where I need to express a double sum in terms of two known series, p and q. Let me first write down what's given. We have: $p = \sum_{k=1}^{\infty} \frac{1}{k^2}$ and $q = \sum_{k=1}^{\infty} \frac{1}{k^3}$. And the expression we need to evaluate is $\sum_{j=1}^{\infty} \sum_{k=1}^{\infty} \frac{1}{(j+k)^3}$. Hmm, so this is a double infinite sum where both j and k start from 1 and go to infinity, and each term is 1 over $(j+k)$ cubed. I need to find a way to write this in terms of p and q. Let me think about how to approach this. Maybe I can find a substitution or change variables to simplify the expression. Since the denominator is $(j+k)^3$, maybe if I let $n = j + k$, then I can rewrite the sum in terms of n and another variable. Let's try that. Let me set $n = j + k$. Then, for each fixed n, j can range from 1 to n - 1, because k would be n - j, which also needs to be at least 1. So, for each n, j goes from 1 to n - 1, and k = n - j. So, substituting, the double sum becomes: $\sum_{n=2}^{\infty} \sum_{j=1}^{n-1} \frac{1}{n^3}$. Wait, that seems promising. Because now for each n, we have a term $\frac{1}{n^3}$ multiplied by the number of j's, which is (n - 1). So, the inner sum is just $\frac{n-1}{n^3}$. Therefore, the entire expression becomes: $\sum_{n=2}^{\infty} \frac{n-1}{n^3}$. Simplify that: $\sum_{n=2}^{\infty} \left( \frac{1}{n^2} - \frac{1}{n^3} \right)$. Which is: $\sum_{n=2}^{\infty} \frac{1}{n^2} - \sum_{n=2}^{\infty} \frac{1}{n^3}$. But I know that: $\sum_{n=1}^{\infty} \frac{1}{n^2} = p$ and $\sum_{n=1}^{\infty} \frac{1}{n^3} = q$. So, subtracting the first term from both series: $\sum_{n=2}^{\infty} \frac{1}{n^2} = p - 1$ and $\sum_{n=2}^{\infty} \frac{1}{n^3} = q - 1$. Therefore, substituting back into the expression: $(p - 1) - (q - 1) = p - 1 - q + 1 = p - q$. Wait, so the entire double sum simplifies to p - q? Let me verify my steps to make sure I didn't make a mistake. 1. I started by changing variables, setting n = j + k. Then for each n, j goes from 1 to n - 1, so k = n - j also goes from 1 to n - 1. That seems correct. 2. Then, I rewrote the double sum as a single sum over n, with each term being $\frac{n-1}{n^3}$. That makes sense because for each n, there are (n - 1) terms contributing $\frac{1}{n^3}$ each. 3. Then, I split the fraction into $\frac{1}{n^2} - \frac{1}{n^3}$. That's a valid step because $\frac{n-1}{n^3} = \frac{n}{n^3} - \frac{1}{n^3} = \frac{1}{n^2} - \frac{1}{n^3}$. 4. Then, I separated the sums into two infinite series starting from n = 2, which is correct because the original sum starts from n = 2. 5. Finally, I expressed the sums in terms of p and q by subtracting the n=1 term from each series. That's correct because the original p and q start from n=1. So, putting it all together...
**[OMITTED]**
Therefore, the final answer is: $\boxed{p - q}$ ✓

Figure 10: Ada-R1 can infer answers to difficult problems with relatively few long CoT, whereas the R1 model requires longer CoT.

