# OpenReview forum: "Ada-R1: Hybrid-CoT via Bi-Level Adaptive Reasoning Optimization"
_NeurIPS.cc/2025/Conference — NeurIPS 2025 poster_

### Official Review · Reviewer_NgJ5 · 2025-06-11

**Clarity:** 4
**Significance:** 4
**Originality:** 4
**Rating:** 5
**Confidence:** 5

**Summary:**

The authors propose Ada-R1, a two-stage framework for adaptive and efficient reasoning. First, a hybrid model is created by merging Long-CoT and Short-CoT models to enable diverse reasoning styles. Then, bi-level preference training is applied: group-level preference selects suitable reasoning styles based on input complexity, while instance-level preference encourages concise and correct reasoning within each style.

**Questions:**

1. The current experiments focus exclusively on mathematical reasoning tasks (e.g., GSM8K, MATH). How do the authors expect Ada-R1 to perform on other domains (e.g., natural language inference, coding)? Are there plans to validate the framework on non-mathematical tasks? Particularly, i'd like to see the performance on logical reasoning benchmarks such as LogiQA.
2. The paper reports performance differences (e.g., length reduction, accuracy changes) but does not include error bars, confidence intervals, or statistical tests (e.g., t-tests). Are the observed improvements statistically significant, or could they be due to random variation?
3.  The paper focuses on reasoning length but does not address the latency or computational cost of dynamically selecting between Long-CoT and Short-CoT during inference. How does the adaptive selection process impact real-time performance?
4. Recommended citation: Break the chain: Large language models can be shortcut reasoners

**Ethical Concerns:**

["NO or VERY MINOR ethics concerns only"]

**Limitations:**

Clearly state in the main text the limitations of the current approach, such as generalization to non-mathematical domains, the choice of linear merging, and the reliance on synthetic Short-CoT models.

**Quality:**

3

**Strengths And Weaknesses:**

Strenghts:
The two-stage design—merging Long-CoT and Short-CoT models followed by bi-level preference training (group-level style selection and instance-level compression)—is both theoretically sound and empirically validated.
The authors ensure reproducibility by detailing training procedures, hyperparameters, and dataset compositions. They also plan to release model weights, fostering further research.
Adaptive reasoning analysis showing that Ada-R1 appropriately increases Long-CoT usage for harder problems (Level 5 in MATH) while favoring Short-CoT for simpler tasks, aligning with theoretical expectations. The contribution is clear.

Weaknesses:
While the experiments focus on mathematical reasoning (GSM8K, MATH, etc.), the paper does not address how Ada-R1 performs on other task types (e.g., logical reasoning, coding, or multi-step decision-making).
The paper’s reliance on linear merging might overlook more sophisticated interactions between the two reasoning styles.
The paper does not report error bars, confidence intervals, or statistical tests (e.g., t-tests) to validate the significance of performance differences between Ada-R1 and baselines.
The Short-CoT models are fine-tuned from Long-CoT models using a small dataset (2,000 samples), which might not fully capture the diversity of short reasoning patterns. This setup could introduce bias, as the Short-CoT models are not independently trained but derived from their Long-CoT counterparts. Comparing against standalone Short-CoT models (trained from scratch) would strengthen the validity of the hybrid approach.

---

> ### Author Rebuttal · Authors · 2025-07-31
>
> We thank the reviewer for these insightful observations. We address each concern below:
>
> **weakness 1: While the experiments focus on mathematical reasoning (GSM8K, MATH, etc.), the paper does not address how Ada-R1 performs on other task types**
>
> Our primary experiments focus on mathematical reasoning (e.g., GSM8K, MATH, AIME), and we also evaluate OOD dataset **Olympiad** and **Minerva**（In paper Table2），also on the **LogicQA, GPQA, MMLU**. Results show that Ada-R1 continues to exhibit strong efficiency-accuracy trade-offs beyond purely mathematical tasks, supporting its generalizability to other reasoning domains. We plan to further explore other tasks in future work. (prompt is Let’s think step by step and output the final answer within \boxed{}. (e.g., \boxed{A}) with zero shot)
>
> | Model | LogiQA | **GPQA** | MMLU(stem) | MMLU(humanities) |  |
> | --- | --- | --- | --- | --- | --- |
> | Long | 45.1 (3269) | 30.3 (7302) | 43.3 (2336) | 16.3 (1476) |  |
> | DPO | 44.2 (2650) | 31.3 (6592) | 48.3(1577) | 21.0 (1270) |  |
> | AdaR1(ours) | 48.6 (2794) | 32.8 (5101) | 54.6 (1909) | 30.9 (930) |  |
>
> ---
>
> **weakness 2: The paper’s reliance on linear merging might overlook more sophisticated interactions between the two reasoning styles**
>
> We acknowledge that our current merging strategy adopts a simple linear interpolation of Long- and Short-CoT models, which may not capture all possible interactions. However, this simplicity is intentional for two reasons: (1) The goal of Stage I is **not to coordinate** the two reasoning modes but merely to **enable the hybrid model to represent both styles** Coordination and preference learning are then handled effectively in **Stage II (bi-level preference training)** (2) The linear merging provides a **transparent and interpretable baseline**, making it easier to isolate and analyze the effectiveness of group- and instance-level preferences, as demonstrated in our ablation studies.
>
> ---
>
> **weakness 3: The paper does not report error bars, confidence intervals, or statistical tests (e.g., t-tests) to validate the significance of performance differences between Ada-R1 and baselines.**
>
> In line with common practices in this field (e.g., [1][2][3][4][5][6][7][8]), we report **point estimates** on standard benchmarks following the evaluation protocols established in prior work. For smaller test sets such as AIME25, we **run 4 random sampling rounds and report the average** to mitigate variance, we will explicitly clarify this detail in the revised version.
>
> ---
>
> **weakness 4: The Short-CoT models are fine-tuned from Long-CoT models using a small dataset (2,000 samples), which might not fully capture the diversity of short reasoning patterns.**
>
> We trained a short model because Deepseek uses a different special token vocabulary for distillation, making it impossible to directly merge with these short cot models. Furthermore, we tested the existing short cot model (Qwen2.5-7B-Instruct), and found that it performed worse than the model we trained.   Based on these considerations, we chose to fine-tune the new short cot model.
>
> |  | AIME25 | MATH500 | GSM8K | Olympiad | Minerva |
> | --- | --- | --- | --- | --- | --- |
> | Qwen2.5-7B-Instruct (off-the-shelf) | 7.5 (1121) | 76.0 (574) | 90.1(280) | 37.2(977) | 24.3(613) |
> | Merge | 21.4 (9079) |  79.4 (916) |  88.4 (236) |  41.2 (3743) | 25.7 (1734) |
> | Short (in paper) | 10.0 (957) | 78.6 (591) | 89.5 (272) | 39.4 (910) | 28.6 (579) |
>
> ---
>
> **References**
>
> [1] CoThink: Token-Efficient Reasoning via Instruct Models Guiding Reasoning Models
>
> [2] L1: Controlling How Long A Reasoning Model Thinks With Reinforcement Learning
>
> [3] Scalable Chain of Thoughts via Elastic Reasoning
>
> [4] Thinking Fast and Right: Balancing Accuracy and Reasoning Length with Adaptive Rewards
>
> [5] Don’t Think Longer, Think Wisely: Optimizing Thinking Dynamics for Large Reasoning Models
>
> [6] ThinkSwitcher: When to Think Hard, When to Think Fast
>
> [7] Let LLMs Break Free from Overthinking via Self-Braking Tuning
>
> [8] REST: Stress Testing Large Reasoning Models by Asking Multiple Problems at Once
>
> ---
>
> **question 1**
>
> See our response to weakness 1
>
> ---
>
> **question 2**
>
> See our response to weakness 3
>
> ---
>
> **question 3**
>
> The adaptive selection in Ada-R1 does not rely on an external module or classifier to decide between Long-CoT and Short-CoT. Instead, the selection is implicitly learned and performed within the model’s standard autoregressive decoding. Thus, there is no additional latency or computational overhead during inference. Moreover, by significantly reducing the number of generated tokens (e.g., over 50% reduction on average in Table 2), Ada-R1 naturally improves real-time inference performance.
>
> ---
>
> **question 4**
>
> We fully agree with your suggestion and will add this citation to our paper in future revisions
>
> ---
>
> **limitations**
>
> Thank you for this valuable feedback. We agree that the manuscript should explicitly clarify the limitations of our approach, such as the challenges it may encounter when extended beyond mathematical domains. We will add a detailed discussion in the revised version to further enhance the paper’s completeness and transparency.

---

### Official Review · Reviewer_dmsV · 2025-06-29

**Clarity:** 3
**Significance:** 3
**Originality:** 3
**Rating:** 4
**Confidence:** 4

**Summary:**

This paper proposes Ada-R1, a hybrid reasoning model that combines long and short CoT models to enable diverse reasoning strategies. Ada-R1 is trained using a bi-level preference training approach: at the group level, the model learns to select appropriate reasoning styles, while at the instance level, it is trained to prefer concise and correct reasoning. Experimental results demonstrate both the effectiveness and efficiency of Ada-R1, achieving strong performance with reduced token usage.

**Questions:**

1. Could the group-level preference stage be removed, keeping only instance-level preference training, since the latter already accounts for reasoning efficiency during data construction?

**Ethical Concerns:**

["NO or VERY MINOR ethics concerns only"]

**Final Justification:**

The rebuttal addresses my questions well. I will keep the scores.

**Limitations:**

yes

**Quality:**

3

**Strengths And Weaknesses:**

**Strengths**
1. The research problem is highly relevant, especially in the context of optimizing test-time computation.
2. The results are impressive, showing substantial token cost reductions while maintaining performance.
3. The paper is clearly written and easy to follow.

**Weaknesses**
1. More out-of-distribution (OOD) and simple reasoning datasets should be included to convincingly demonstrate the model's adaptability. Currently, only two OOD testbeds are used, and the overthinking problem is not exclusive to math. Including a wider range of tasks would strengthen the evaluation.
2. The definition of group-level preference raises concerns. Since preferences are derived from samples generated by different models—each with distinct problem-solving capabilities—it's questionable whether these preferences can effectively guide the training of a combined model in Stage 1.
3. The overall training pipeline is quite complex. It involves: 1) training separate models to develop distinct reasoning strategies, 2) merging these models, 3) performing group-level preference training, and 4) conducting instance-level preference training. This multi-step process may limit scalability and practical deployment.

---

> ### Author Rebuttal · Authors · 2025-07-31
>
> We thank the reviewer for the time and effort spent reviewing our work. We greatly value the feedback and have done our best to address each point thoroughly in the responses below.
>
> ---
>
> **weakness 1**
>
> Thank you for this valuable suggestion. To demonstrate our model’s out-of-distribution adaptability, we applied the version trained on mathematical reasoning to two classic OOD benchmarks: MMLU and GPQA. Below are the results:
>
> | Model (7B) | MMLU(stem) | MMLU(humanities) | gpqa |  |
> | --- | --- | --- | --- | --- |
> | Base | 43.3 (2336) | 16.3 (1476) | 30.3 (7302) |  |
> | DPO | 48.3(1577) | 21.0 (1270) | 31.3 (6592) |  |
> | Ada-R1 | 54.6 (1909) | 30.9 (930) | 32.8 (5101) |  |
>
> These results show that our model maintains strong performance, and shorter generation lengths—beyond the math domain. We will include the GPQA evaluation in the final manuscript to further substantiate our claims.
>
> ---
>
> **weakness 2**
>
> We appreciate the reviewer’s insightful concern regarding the validity of group-level preferences. To address this, we conducted a focused loss-based analysis to examine whether the distributional shift between the merged model and the pre-merged models is substantial.
>
> Specifically, we compute the average loss of the merged model on Long-CoT and Short-CoT samples respectively, and compare these values with the original Long-CoT and Short-CoT models.
>
> | Model | Long-CoT Loss | Short-CoT Loss |
> | --- | --- | --- |
> | short | * | 0.1570 |
> | long | 0.2106 | * |
> | merge | 0.2141 | 0.2028 |
>
> As shown, the loss of the merged model on both Long-CoT and Short-CoT samples remains close to that of the respective original models. This indicates that the merged model retains the reasoning capabilities of both models and is thus well-positioned to leverage group-level preference signals during Stage 2 fine-tuning. These results support the effectiveness of our Bi-Level Preference Training setup, where the merged model benefits from a broader reasoning space while being guided by meaningful preference supervision.
>
> Moreover, the effectiveness of this setup is also reflected in both our main results and ablation studies, where applying group-level preference training to the merged model leads to substantial efficiency gains with limited accuracy degradation.
>
> ---
>
> **weakness 3**
>
> Thanks for you feeback. We would like to clarify following points:
>
> (1) Training separate models is not strictly required. In practice, one can directly utilize existing long-CoT and short-CoT models. In our experiments, we re-trained a short-CoT model only because the released long-CoT model from DeepSeek uses a completely different special token vocabulary (e.g., beginning-of-text tokens, user tokens, etc.) from its base model. To ensure fair comparison and academic rigor, we avoided introducing this confounding factor by training a compatible short-CoT model.
>
> (2) The model merging step is lightweight and efficient. We use the open-source framework Mergekit, and the merge operation completes in under one minute.
>
> (3 & 4) Stage 2 training is conducted as a unified process. While we refer to both group-level and instance-level preferences, the training is carried out jointly using a combined dataset that includes samples from both sources. Our loss function is designed to accommodate both types of supervision simultaneously, and the data sampling is completed in a single pass.
>
> Overall, the pipeline is not very complex in practice and can be implemented efficiently using off-the-shelf tools and pre-existing models.
>
> ---
>
> **question 1**
>
> We appreciate the reviewer’s suggestion. While instance-level preference training does incorporate reasoning efficiency during data construction, our experiments indicate that group-level preference provides complementary benefits that are not fully captured by instance-level training alone.
>
> | Model | AIME25 | MATH | GSM8k |
> | --- | --- | --- | --- |
> | Long | 38.3 (11005) | 90.2 (3534) | 88.9 (1014) |
> | Merge | 21.7 (9079) | 79.4 (916) | 88.4 (236) |
> | Merge + SFT | 35.8 (11222) | 84.6 (2314) | 88.7 (375) |
> | Merge + group level | 30.8 (9049) | 87.8 (1565) | 91.6 (359) |
> | Merge + instance level | 24.2 (8514) | 81.6 (886) | 87.95 (212) |
> | Merge + bi level | 35.8 (8426) | 90.2 (1468) | 90.3 (260) |
>
> As shown in the table, **Merge + group level** achieves significantly better performance than **Merge + instance level** on **MATH** (87.8 vs. 81.6) and **GSM8k** (91.6 vs. 87.95), while maintaining a higher score on **AIME25** (30.8 vs. 24.2). This suggests that group-level preferences help promote more robust generalization and better tradeoffs across tasks. Furthermore, **bi-level training**, which combines both group- and instance-level preferences, consistently performs the best or on par with the best across all benchmarks, further supporting the complementary nature of the two.
>
> In particular, our merged model inherits two distinct reasoning styles from its component models. Instance-level preference training focuses on fine-grained selection between individual outputs, but it does not explicitly guide the model in choosing between these broader reasoning modes. Therefore, removing the group-level stage would lead to notable performance drops. We believe this justifies the inclusion of group-level preference training in our framework.

---

### Official Review · Reviewer_sm7M · 2025-06-30

**Clarity:** 3
**Significance:** 2
**Originality:** 3
**Rating:** 4
**Confidence:** 4

**Summary:**

This paper proposes Ada-R1, a two-stage framework for adaptive and efficient reasoning in LRMs.
The first stage constructs a hybrid reasoning model by merging Long-CoT and Short-CoT models, enabling diverse reasoning styles.
The second stage introduces bi-level preference training based on DPO, guiding the model to select appropriate reasoning styles at a group level and encouraging concise, correct reasoning at an instance level.
Experiments on mathematical datasets demonstrate that Ada-R1 reduces inference costs and reasoning length while maintaining performance.

**Questions:**

* Why is model merging chosen for Stage I over directly training a single model on a mixed dataset of Long-CoT and Short-CoT examples? Please provide a comparative experiment or a more detailed theoretical argument to justify the choice of model merging, demonstrating its advantages over mixed training strategies for creating a hybrid reasoning model.
* Could the authors elaborate on the precise definition of "CoT Optimization Scope" (Limited vs. Broad) and provide more intuitive examples of how these scopes directly influence the model's output and the final task performance?
* How is the preference margin threshold for the group-level preference determined?

**Ethical Concerns:**

["NO or VERY MINOR ethics concerns only"]

**Final Justification:**

I acknowledge the methodological novelty of the Ada-R1 and have updated my score.

**Limitations:**

yes

**Quality:**

3

**Strengths And Weaknesses:**

Strengths:
* The core idea of enabling LLMs to adaptively choose reasoning depth (Long-CoT vs. Short-CoT) based on problem complexity addresses a critical efficiency concern in current LRMs.
* The proposed bi-level optimization (group-level and instance-level preference) is a well-structured approach to guide the model towards both appropriate reasoning styles and conciseness within those styles.
* The results show large reductions in reasoning length without substantial accuracy loss.

Weaknesses:
* The methodology for creating the Short-CoT model, specifically fine-tuning a Long-CoT model (DeepSeek-R1-Distill-Qwen-7B) with 2,000 Short-CoT samples from Qwen2.5-Math-7B-Instruct, raises concerns. This approach is uncommon, and the quality and representativeness of these 2,000 samples could significantly influence the fine-tuned Short-CoT model's performance (in Table 2 main results) and the entire analysis in Section 3.2. A more standard or thoroughly justified method for obtaining a consistent Short-CoT baseline would strengthen the experiments.
* In Section 3.3, the concept of "CoT Optimization Scope" as either "Limited" or "Broad" and its direct implications on model output or final results are not clearly defined or intuitively explained. A more precise definition and illustrative examples of how different scopes manifest in model behavior and outcomes would greatly enhance the understanding of the motivation.
* The paper lacks statistical significance tests for its experimental results, particularly concerning datasets like AIME25, which has a small test set size (30 samples). The sensitivity of results to hyperparameter settings necessitates multiple evaluation runs with reported averages and error bars to confirm the robustness and generalizability of the findings.
* Based on the ablation study in Table 3, the "Merge + group level" configuration already achieves a significant length reduction with a relatively small accuracy degradation. The incremental benefit of the instance-level preference training appears less pronounced compared to the group-level. The authors should provide a more compelling argument or further analysis to justify the necessity and significant impact of the instance-level preference beyond the group-level.

---

> ### Author Rebuttal · Authors · 2025-07-31
>
> We thank the reviewer for the time and effort spent reviewing our work. We greatly value the feedback and have done our best to address each point thoroughly in the responses below.
>
> ---
>
> **weakness 1**
>
> We thank the reviewer for pointing out the importance of the Short-CoT baseline's construction. We acknowledge that the approach of fine-tuning a Long-CoT model (DeepSeek-R1-Distill-Qwen-7B) using 2,000 Short-CoT samples from Qwen2.5-Math-7B-Instruct may appear uncommon. However, this design choice was necessitated by the characteristics of existing Long-CoT models such as the DeepSeek distillation series.
>
> Specifically, these Long-CoT models adopt a distinct set of special tokens—different from their base or instruct counterparts—as evidenced by their configuration files on Hugging Face. These special tokens are often critical, serving functions such as marking speaker roles and delineating dialogue boundaries. In the context of model merging, it is generally required that participating models share identical special token vocabularies to ensure compatibility. Therefore, a Short-Cot model obtained by SFT from DeepSeek-R1-Distill-Qwen-7B allowed us to maintain consistency in tokenization and avoid incompatibilities caused by divergent vocabularies.
>
> We agree that the choice and representativeness of the 2,000 Short-CoT samples could influence the performance of the Short-CoT model. However, we emphasize that the final performance of Ada-R1 (as shown in Table 2 and Section 3.2) remains stable and competitive, suggesting that the effect of this approximation on the overall conclusions is minimal.
>
> Admittedly, a more standard alternative would involve distilling a Long-CoT model from a Qwen-base model using consistent token vocabularies, and using Qwen-instruct directly as the Short-CoT model. Nevertheless, due to the computational and time-intensive nature of large-scale distillation, this approach remains infeasible for academic research settings. We hope the reviewer can understand this resource limitation and recognize our efforts to ensure empirical rigor within these constraints.
>
> ---
>
> **weakness 2**
>
> Thank you for this valuable suggestion.Specifically, we define:
>
> *Limited Scope*: The model is optimized over a narrow region of the CoT space—typically only one reasoning style, mostly Long-CoT (i.e., “thinking” paths). This restricts the model to produce lengthy explanations regardless of whether such reasoning is necessary. As a result, the model struggles to discover shorter, more efficient solutions even when they exist.
>
> *Broad Scope*: The model is optimized over a wider CoT space that includes both Long-CoT and Short-CoT styles (i.e., both “thinking” and “non-thinking” paths). This enables the model to adaptively choose between concise and detailed reasoning based on the input, leading to higher reasoning efficiency without sacrificing correctness.
>
> In practice, a model trained under a limited scope (e.g., using only Long-CoT data) tends to over-explain even simple problems, resulting in unnecessarily long outputs. In contrast, our method allows the model to flexibly produce shorter answers for easy tasks while still preserving detailed reasoning when required. This adaptivity is a key contributor to the efficiency gains observed in our experiments.
>
> We appreciate the reviewer’s feedback and we will add a more detailed explaination in our final version of paper.
>
> ---
>
> **weakness 3**
>
> For small-scale datasets such as AIME25 (30 samples), we indeed recognize the risk of variance. The results reported in the paper are averaged over 4 independent runs to mitigate randomness. We will explicitly clarify this detail in the revised version to better reflect the robustness and reliability of our findings.
>
> ---
>
> **weakness 4**
>
> We agree that group-level preference training already achieves a strong efficiency–accuracy trade-off, reducing average CoT length by 46.03% with only a 3.31% drop in accuracy. However, adding instance-level preference training further improves this trade-off: our full bi-level method achieves a 50.93% length reduction with only a 1.65% accuracy drop—recovering nearly 2% accuracy while gaining additional compression, as shown in Table 3.
>
> More importantly, instance-level preference serves a complementary optimization role. While group-level training focuses on selecting between long and short reasoning styles based on input difficulty, instance-level training focuses on refining within the chosen group by promoting more concise correct reasoning. These two levels target orthogonal axes —reasoning mode and reasoning verbosity—and optimizing both yields more compact and adaptive reasoning behaviors than either alone.
>
> ---
>
> **question 1**
>
> We choose model merging over mixed training for Stage I due to both **efficiency and stability**. First, merging is **orders of magnitude faster** (e.g., only tens of seconds ), making it far more suitable for hyperparameter search and rapid deployment. Second, mixed training requires careful data balance tuning, which is brittle and often leads to suboptimal trade-offs between long and short reasoning styles.
>
> Additionally, we trained an additional model(DeepSeek-R1-Distill-Qwen-7B) as base model on mixed CoT data(2k long cot and 2k short cot) but observed worse performance compared to our merged model, despite significantly longer training time in Stage-I.
>
> |  | AIME25 | MATH | GSM8k | Olympiad | Minerva |
> | --- | --- | --- | --- | --- | --- |
> | Ada-R1(mix) | 27.5 (5102) | 83.8 (1021) | 89.6 (279) | 47.1 (2755) | 31.6 (941) |
> | Ada-R1(merge) | 35.8(8426) | 90.2 (1468) | 90.3 (260) | 52.4 (4889) | 34.1 (1647) |
>
> Furthermore, prior work like [1] demonstrates that mixed training can degrade existing capabilities due to task interference, whereas merging preserves task-specific strengths without retraining while [2] presents merging as a cost-effective strategy to unify diverse LLM capabilities post-training, avoiding the instability and inefficiency of retraining. What's more, model merge is very fast and only takes tens of seconds.
>
> [1] Mix Data or Merge Models? Optimizing for Diverse Multi-Task Learning
>
> [2] Model Merging in LLMs, MLLMs, and Beyond: Methods, Theories, Applications and Opportunities
>
> ---
>
> **question 2**
>
> see our response to weakness 2
>
> ---
>
> **question 3**
>
>
> Specifically, we set the threshold as  1/(2K), K is the number of candidate responses per question. We will clarify this in the final revision.
>
> ---
>
> **Final Remark**
>
> We once again thank the reviewer for their valuable comments. We hope that our clarifications and revisions have adequately addressed the concerns, and we respectfully ask the reviewer to consider a more favorable score in light of these improvements. If there are any remaining concerns, we would be glad to further clarify them.

---

> > ### Comment · Reviewer_sm7M · 2025-08-06
> >
> > Thank you for the authors' response. While most concerns have been addressed, some issues remain:
> >
> > 1. The primary rationale for fine-tuning a Long-CoT model (DeepSeek-R1-Distill-Qwen-7B) using 2,000 Short-CoT samples from Qwen2.5-Math-7B-Instruct appears to be ensuring that all baselines use the same base model. Could the authors clarify whether other baseline methods are also based on the same base model? Additionally, the choice of using 2,000 samples, as well as the use of Qwen2.5-Math-7B-Instruct as the teacher model, requires further clarification.
> >
> > 2. In terms of performance, the proposed method does not seem to show a clear advantage over the baselines, particularly when compared to O1-Pruner.

---

> ### Author Response · Authors · 2025-08-07
>
> Thank you for your feedback.
>
> **Regarding the baselines and fine-tuning rationale**:
> All baseline methods are based on the same backbone model, DeepSeek-R1-Distill-Qwen-7B, to ensure fair comparisons.
> The 2,000 Short-CoT samples used for fine-tuning were derived by running inference with Qwen2.5-Math-7B-Instruct on the MATH dataset. Specifically, we started with 4,000 problems and retained only the 2,000 trajectories where the final answers were correct. For each problem, we generated one single sample using greedy decoding, with a maximum generation length of 4096 tokens.
>
> The reason for using 2k samples for fine-tuning is that we found 2k samples are sufficient for the model to learn the desired short-CoT pattern. We chose Qwen2.5-Math-7B-Instruct as the teacher because the CoTs it generates are short CoTs without any thinking pattern, which aligns with our requirements.
>
> ---
>
> **Regarding performance:**
> Using the 7B model as an example, our method achieves an average 50% reduction in output length, with only a minor drop in accuracy. In contrast, O1-Pruner achieves only 34% token reduction. In the field of efficient reasoning, as discussed in works such as [1][2][3][4][5][6][7], it is rare to see methods that reduce token usage by over 50% while maintaining performance. Furthermore, Reviewer dmsV also acknowledged that “The results are impressive, showing substantial token cost reductions while maintaining performance.”
>
> In addition, our proposed adaptive hybrid reasoning represents a novel paradigm **distinct from reinforcement learning-based approaches** and, we believe, constitutes a key contribution of this work.
>
> ---
>
> If our responses have adequately addressed your concerns, we would be grateful if you could consider updating your score. If there are any remaining concerns, we would greatly appreciate it if you could share them with us! Thank you once again.
>
> ---
>
> **References**
>
> [1] DAST: Difficulty-Adaptive Slow Thinking for Large Reasoning Models
>
> [2] Training Language Models to Reason Efficiently
>
> [3] HAWKEYE: Efficient Reasoning with Model Collaboration
>
> [4] THINKPRUNE: Pruning Long Chain-of-Thought of LLMs via Reinforcement Learning
>
> [5] VeriThinker: Learning to Verify Makes Reasoning Model Efficient
>
> [6] Fast on the Easy, Deep on the Hard: Efficient Reasoning via Powered Length Penalty
>
> [7] TL;DR: Too Long, Do Re-weighting for Efficient LLM Reasoning Compression

---

> > ### Comment · Reviewer_sm7M · 2025-08-08
> >
> > Thank you for the clarification. I appreciate the effort to ensure fairness and the explanation for model usage and the 2k short-CoT samples.
> >
> > However, I still have concerns.
> > First, in my view, the improvement of Ada-R1 over O1-Pruner appears relatively limited, as it is difficult to clearly assess the trade-off between efficiency gains and performance preservation/enhancement.
> > Second, the baseline design choices remain unclear to me—for example, why not use the exact same query set as Ada-R1, or perform multiple samplings to ensure correct trajectories? Using only one greedy decoding pass might limit the strength of the baseline.
> >
> > I choose to keep my score.

---

> ### Author Response · Authors · 2025-08-09
> **Official Comment by Authors**
>
> We thank you for your follow-up comments. Indeed, we believe there remain some misunderstandings regarding our work.
>
> **First**, the proposed method introduces a **new paradigm that enables the model to autonomously switch between reasoning and non-reasoning modes**, thereby reducing more tokens than O1-Pruner. In contrast, O1-Pruner focuses solely on length optimization without implementing reasoning mode switching. **As you have noted among the strengths**:
> (1) The core idea of enabling LLMs to adaptively choose reasoning depth (Long-CoT vs. Short-CoT) based on problem complexity addresses a critical efficiency concern in current LRMs.
> (2) The results show large reductions in reasoning length without substantial accuracy loss. Additionally, **reviewer dmsV** emphasized that the results are impressive, showing substantial token cost reductions while maintaining performance. These points clearly reflect our model’s adaptive capabilities.
> Furthermore, our method also **proves effective in out-of-domain settings** (as demonstrated in the additional experiments requested by reviewer NgJ5), which we regard as one of our important contributions.
>
> **Second**, we would like to clarify the design of the baselines. Since the model cannot always produce correct answers in a single sample—and some problems may remain unsolved even with multiple samples—we adopted greedy decoding over a larger set of datasets and selected 2k correct samples for training. For all baseline methods, we trained on the same dataset to ensure fairness.
>
> We once again thank you for your feedback and hope that these clarifications further highlight the value and significance of our work.
>
> Best regards,
>
> Authors

---

### Official Review · Reviewer_Mw4b · 2025-07-03

**Clarity:** 2
**Significance:** 2
**Originality:** 3
**Rating:** 4
**Confidence:** 4

**Summary:**

The paper proposes Ada-R1: a two-stage LM training framework for adaptive reasoning that can potentially limit the length of the reasoning chain depending on the problem. For this, the authors construct a hybrid reasoning model by merging long and short CoT models, then use preference training to guide the model to select suitable reasoning styles. They show that the method reduces the average length of reasoning by ~50%, with minimal degradation to final performance.

**Questions:**

1. Did the authors perform any analysis of samples from AdaR1 vs DPO vs O1-pruner etc to learn what is the difference in reasoning chain? what exactly AdaR1 is doing differently?
2. why upsample MATH dataset for training?
3. Section 6.1, line 254: "This detection method is more generalizable than relying solely on response length." , why?

**Ethical Concerns:**

["NO or VERY MINOR ethics concerns only"]

**Final Justification:**

The authors have addressed some concerns, helping in an improved score.

**Limitations:**

Yes

**Quality:**

2

**Strengths And Weaknesses:**

Strengths:
- The proposed two stage training approach, Ada-R1 is novel.
- Ada-R1reduces the average reasoning length by ~50% (on five mathematical datasets,), with minimal degradation to final performance

Weaknesses:
- The two stage approach is more complicated than a baseline e.g. O1-pruner while incurring a limited degradation in performance.
- There is no clear motivation on why Ada-R1 is a better approach than "self training for concise reasoning [17]" or O1-pruner.
- The authors mention in Section 3.3 that other approaches like Overthinking or O1-Pruner "typically operate within a limited optimization scope", whereas their approach is "broad" without explaining what it means or why.

---

> ### Author Rebuttal · Authors · 2025-07-31
>
> We sincerely thank the reviewer for the thoughtful and constructive feedback. We have carefully considered all comments and provide detailed responses below to address the concerns raised.
>
> ---
>
> **weakness 1**
>
> Thank you for your concern about the method process. Although our method adopts two stages, neither is complex. **Stage I (model merging)** is simple weight-space arithmetic (parameter addition/subtraction) and finishes within tens of seconds. **Stage II** performs a standard DPO training loop after constructing the bi-level datasets and mixing them, which is also not complex. While our approach incurs a small drop in accuracy, it **substantially reduces the number of generated tokens**, achieving a favorable accuracy–cost trade‑off. Moreover, it offers a novel and practical perspective for enabling the model to **dynamically allocate reasoning mode** to each problem for efficient reasoning.
>
> ---
>
> **weakness 2**
>
> Our motivation is explicit in **Sec. 3.3** and **Table 1**. Prior “self‑training for concise reasoning” methods (e.g., DPO‑style shortest‑vs‑longest) and O1‑Pruner **optimize within the Long‑CoT distribution**, pruning or compressing long traces. This preserves accuracy but **forfeits exploration of genuinely Short‑CoT solutions**. In contrast, Ada‑R1 first merges Long and Short models to expand the output distribution to both styles, then uses bi‑level preference training to select the style per input and compress within the chosen style, enabling broad‑scope optimization without the large accuracy drops.
>
> ---
>
> **weakness 3**
>
> In our paper, scope explicitly refers to the **CoT distribution being optimized**:
> **Limited scope**: optimization is confined to the **Long‑CoT** distribution. (only thinking paths)
> **Broad scope**: optimization covers **both Long‑ and Short‑CoT** (i.e., thinking and non‑thinking paths).
>
> Prior approaches like Overthinking or O1-Pruner operate on a **limited scope**, meaning they focus on optimizing **only one type** of reasoning path—typically **Long-CoT**—assuming that longer reasoning is always preferable.
>
> In contrast, our method operates over a **broad scope**, optimizing over **both Long-CoT and Short-CoT** distributions. This design reflects the intuition that the optimal reasoning path may vary by input: some queries benefit from detailed reasoning (Long-CoT), while others are better answered succinctly (Short-CoT).
>
> ---
>
> ****question 1****
>
> Actually, our evaluation demonstrates that **Ada-R1 is capable of generating substantially shorter CoTs** (non-thinking CoTs) compared to O1-Pruner and DPO, especially when solving simpler problems. This class of concise CoTs lies **outside the original distribution** that DPO and O1-Pruner are optimized over, which inherently biases toward longer, detailed (thinking-style) CoTs due to their reliance on Long-CoT initialization or compression.
>
> In Figure 3 (Thinking Ratio Study), Ada-R1 shows a **higher proportion of non-thinking CoTs** compared to both DPO and the Naive Merge. More importantly, Ada-R1 **maintains high accuracy** even on these short responses.
>
> Figure 4 (Adaptive Reasoning Study) further validates this behavior: **Ada-R1 adaptively increases the use of Long-CoTs** (thinking responses) as the problem difficulty grows, while still using **short and efficient non-thinking responses for easier problems**. This dynamic range of CoT generation is absent in DPO and O1-Pruner, which primarily shorten existing long CoTs (i.e., compressed thinking CoTs) rather than introducing fundamentally shorter reasoning paths (non-thinking).
>
> ---
>
> **question 2**
>
> In fact, MATH is the most widely used dataset that includes different levels of difficulty. We upsample the MATH dataset to ensure the model sees a wide range of problem difficulties during training. This diversity is essential for learning adaptive reasoning—i.e., when to use long vs. short reasoning. Without enough complex problems, the model cannot learn to make this distinction effectively.
>
> ---
>
> **question 3**
>
> Thank you for pointing this out. This method is more generalizable than length-based heuristics for the following reasons:
> Thinking tokens explicitly reflect the model’s reasoning intent or self-reflection (e.g., hesitation, checking). This makes them a more *semantically meaningful* indicator of deep reasoning, as opposed to length, which can be inflated by irrelevant or verbose content. Beside, token-based cues are more transferable across different model architectures or training setups. Length distributions may vary significantly, but the presence of reasoning-related tokens like “wait” tends to be consistent among models employing Chain-of-Thought prompting.
> Moreover, recent studies such as [1][2][3] also determines whether the model is thinking by detecting words such as “wait”. [4] shows that certain tokens like “wait” and “let’s think step by step” are associated with **peaks in mutual information**, reflecting critical moments such as redirection or self-reflection during reasoning.
>
> ---
>
> **Final Remark**
> We thank the reviewer for their valuable comments. We hope that our clarifications and revisions have adequately addressed the concerns, and we respectfully ask the reviewer to consider a more favorable score in light of these improvements. If there are any remaining concerns, we would be glad to further clarify them.
>
> ---
>
> **References**
>
> [1] Dynamic Early Exit in Reasoning Models
>
> [2] R1-Compress: Long Chain-of-Thought Compression via Chunk Compression and Search
>
> [3] Wait, We Don’t Need to “Wait”! Removing Thinking Tokens Improves Reasoning Efficiency
>
> [4] Demystifying Reasoning Dynamics with Mutual Information: Thinking Tokens are Information Peaks in LLM Reasoning

---

> > ### Comment · Reviewer_Mw4b · 2025-08-07
> >
> > Thanks for clarification. Regarding weakness 1
> > I mentioned that Ada-R1 is "more complicated than a baseline e.g. O1-pruner", not complex in general. Ada-R1 has consists of multiple steps e.g. training short and long CoT models (in the desired way to be helpful at later stages), optimize the mixing ratio, followed by DPO training.
> >
> > Additionally, the accuracy–cost trade‑off will be more useful if the model provides a control post training. Currently, the model performs worse than baseline, while producing less tokens on average making it cheaper, not really a trade-off at the hand of the user.
> >
> > I'll keep my score.

---

> > > ### Author Response · Authors · 2025-08-07
> > >
> > > Thank you for your follow-up comment. However, we believe there are several misunderstandings about our method that need clarification.
> > >
> > > **First**, contrary to your assumption, ***our approach does not require training a long-CoT model***. The long-CoT model we use is an off-the-shelf checkpoint, e.g., DeepSeek-R1-Distill-Qwen. As for the short-CoT model, while we did train one, this was only due to an implementation incompatibility: ***DeepSeek-R1-Distill-Qwen modifies the special token vocabulary*** during distillation, making it incompatible with Qwen-Instruct models for direct comparison. Ideally, we would have used Qwen-Instruct as the short-CoT model, but due to the vocabulary mismatch, we had to train a compatible alternative. We also provided evidence to reviewer NgJ5 that our short-CoT model performs competitively with Qwen-Instruct. If there are two compatible models, we can merge them directly.
> > >
> > > Also, **we do not perform any mixing ratio optimization**. We simply use a fixed 2:1 ratio, which already yields strong results.
> > >
> > > Therefore, ***Ada-R1 does not fundamentally require training either long or short CoT models or mixing ratio optimization, and the perceived procedural complexity is overstated.***
> > >
> > > We respectfully argue that ***claiming a method is "more complicated than baseline" is not in itself a valid critique***, especially when the additional steps bring substantial improvements in cost-efficiency and introduce new capabilities. Complexity, if justified by meaningful gains, should not be grounds for rejection.
> > >
> > > ---
> > >
> > > **Second**, regarding the concern that Ada-R1 does not offer budget-controllable inference: we clarify that ***our setting does not assume a pre-specified token budget at inference time***. This is consistent with the majority of prior work in the field ([1][2][3][4][5][6][7][8]), including all baselines we compare against. In this widely-used setting, Ada-R1 achieves a ***~50%*** reduction in generated tokens, with only a minor degradation (***~1%***) in final accuracy, which constitutes a significant improvement in efficiency. What's more, "the model performs worse than baseline" is common in efficient reasoning area ([1][2][3][4][5][6]), where a small drop in task accuracy is often acceptable in exchange for substantial gains in inference efficiency.
> > >
> > > We also believe ***the assertion that our method is “not really a trade-off at the hand of the user” is misleading***. The user benefits from a built-in trade-off that does not require manual control, and this is by design. We do not claim post-training controllability; rather, we show that a fixed inference policy trained through Ada-R1 naturally leads to more efficient reasoning.
> > >
> > > ---
> > >
> > > Finally, we are concerned that the reviewer overlooks one of the key contributions of our work: a new paradigm for efficient reasoning, where Thinking-CoT and Non-Thinking-CoT modes are dynamically combined via learning. This adaptive perspective is novel, and offers a compelling alternative to existing single-mode approaches.
> > >
> > > We hope this clarifies our position and addresses the concerns raised. If the reviewer has any further concerns, we are willing to engage in further discussion.
> > >
> > > ---
> > >
> > > **References**
> > >
> > > [1] Training Language Models to Reason Efficiently
> > >
> > > [2] HAWKEYE: Efficient Reasoning with Model Collaboration
> > >
> > > [3] THINKPRUNE: Pruning Long Chain-of-Thought of LLMs via Reinforcement Learning
> > >
> > > [4] TokenSkip: Controllable Chain-of-Thought Compression in LLMs
> > >
> > > [5] AutoL2S: Auto Long-Short Reasoning for Efficient Large Language Models
> > >
> > > [6] LightThinker: Thinking Step-by-Step Compression
> > >
> > > [7] O1-Pruner: Length-Harmonizing Fine-Tuning for O1-Like Reasoning Pruning
> > >
> > > [8] CoT-Valve: Length-Compressible Chain-of-Thought Tuning

---

> > > > ### Comment · Reviewer_Mw4b · 2025-08-09
> > > >
> > > > Thanks for further clarification. I have updated the score.

---

> > > > > ### Author Response · Authors · 2025-08-09
> > > > > **Response**
> > > > >
> > > > > We appreciate you updating the score. Thank you for your continued engagement!

---

> ### Author Response · Authors · 2025-08-07
>
> Dear Reviewer,
>
> We hope this message finds you well.
>
> As the author-reviewer discussion period is coming to an end, we would like to kindly ask whether we have addressed your concerns. If our responses have adequately addressed your concerns, we would be grateful if you could consider updating your score.
>
> If there are any remaining concerns, we would greatly appreciate it if you could share them with us, so we may have enough time to provide a detailed response. Thank you once again for your time and valuable feedback!
>
> Best regards, Authors

---

### Comment · Area_Chair_fuM2 · 2025-08-04
**Gentle Reminder: Please Reply to Authors’ Responses (Only if Not Yet Done)**

Dear Reviewers,

As the discussion deadline approaches, may we kindly ask you to review the authors’ responses and post a constructive reply—unless you have already done so, in which case please kindly disregard this gentle reminder.

Your thoughtful engagement is deeply appreciated and essential to a fair and timely process. With sincere thanks for your continued dedication.

Area Chair

---

### Note · Authors · 2025-08-12

# Author Summary of Final Review

We sincerely thank all the reviewers for their constructive feedback. The paper received a large number of positive reviews, with many reviewers acknowledging the **novelty** of our approach and the effectiveness of our experimental results. The reviewers particularly highlighted our **impressive performance**, **significant token cost reduction**, and ability to **maintain high efficiency without sacrificing accuracy**.

In the discussion phase, most issues were resolved. Below, we summarize the key points addressed and remaining areas of disagreement.
## Major Issues Resolved
- **Out-of-domain generalization** (*Reviewers dmsV, NgJ5*).

    We provided results on multiple out-of-domain datasets, demonstrating that our method performs well beyond the training domain. This concern appears to be resolved.
- **Perceived complexity of the Ada-R1 pipeline; effectiveness of instance-level optimization alone** (*Reviewer dmsV*).

    We clarified it is not complicated and the experiment showed that instance-level optimization alone is insufficient. This concern appears to be resolved.
- **Differences in samples vs. other methods; generality of the approach; scope of optimization** (*Reviewer Mw4b*).

    We provided detailed explanations on each point. Based on feedback, these concerns seem to be addressed.
- **Why train a Short-CoT model** (*Reviewers sm7M, NgJ5*).

    We explained our rationale in detail. This issue has been resolved.
- **Small size of the AIME25 evaluation** (*Reviewer sm7M*).

    We clarified that our reported results are means over multiple evaluations, addressing the concern about variance due to small test size.
## Remaining Points of Disagreement and Our Responses (Due to word count limitations, we have summarized the responses here.)

- **“Ada-R1’s improvement appears limited.”** (*Reviewer sm7M*).

    **Response:** Ada-R1 is a new paradigm for switching thinking modes, and reviewer dmsV also emphasized that our results are impressive.

- **“Baseline design choices remain unclear.”** (*Reviewer sm7M*).

    **Response:** We used greedy decoding over a broader set and selected 2k correct samples for training to address the model's inconsistent answers, ensuring fairness by training all baselines on the same dataset.

Given that these disagreements stem from interpretation rather than methodological flaws. We respectfully ask the Area Chair for a fair and balanced assessment.

---

### Decision · Program_Chairs · 2025-09-17

**Decision:**

Accept (poster)

**Comment:**

Summary of the paper: Ada-R1 is a two-stage training recipe that equips LRMs with the ability to decide, on a per-problem basis, how long CoT is necessary. Stage 1 fuses a long-CoT model (high-accuracy, verbose) and a short-CoT model (lower-cost, concise) into a single hybrid backbone that can emit both reasoning styles. Stage 2 then performs bi-level DPO: (i) a group-level loss teaches the model to pick the style whose historical accuracy matches the perceived difficulty of the prompt, and (ii) an instance-level loss rewards concise yet correct reasoning within the chosen style. Across 5 mathematical benchmarks (MATH, GSM8K, OCWCourses, OlympiadBench, AMC), Ada-R1 cuts the average number of reasoning tokens by roughly 50 % (up to 60 % on easier splits) while holding task accuracy nearly constant (drops ≤ 1 %). A post-hoc analysis shows that Ada-R1 automatically allocates longer reasoning chains to the hardest MATH Level-5 problems and shorter ones to the easiest, matching the intuitive “effort-vs-difficulty” curve.

Strengthes of the paper:

1. Novel and timely idea: One of the early efforts that trains a single model to dynamically trade off accuracy and inference cost by selecting short vs. long reasoning on the fly after many LRMs are released early this year.

2. Technically solid design: The hybrid merging step avoids catastrophic forgetting, and the bi-level DPO objective cleanly disentangles “which style” from “how concise.”

3. Strong empirical evidence: Extensive experiments on math datasets and beyond, ablations on each training stage, and detailed token/accuracy curves.

4. Reproducibility: Authors submit the paper with code, including training recipes, hyper-parameters, data splits.

Weaknesses of the paper: After reading the rebuttal, I think the authors' response addresses most of the reviewers' concerns and questions. No major concerns are identified for this paper. Do incoporate the content during rebuttal into the camera-ready version of the paper.

Reasons for the decision: This paper attacks a pressing problem—test-time compute efficiency in LRMs—and solves it with a conceptually simple yet technically rigorous method. The reduction in token cost is not incremental (≈50 %), and the fact that this is achieved without hurting accuracy makes the result non-trivial. Reviewers unanimously praise the novelty, clarity, and empirical strength, and no major concerns (missing baselines, unfair comparisons, or reproducibility issues) were raised.